# Unbalanced dendritic inhibition of CA1 neurons drives spatial-memory deficits in the Ts2Cje Down syndrome model

Sergio Valbuena [1], Álvaro García[1], Wilfrid Mazier [1,2], Ana V. Paternain[1] & Juan Lerma[1]*

Overinhibition is assumed one of the main causes of cognitive deficits (e.g. memory impairment) in mouse models of Down syndrome (DS). Yet the mechanisms that drive such exaggerated synaptic inhibition and their behavioral effects remain unclear. Here we report the existence of bidirectional alterations to the synaptic inhibition on CA1 pyramidal cells in the Ts2Cje mouse model of DS which are associated to impaired spatial memory. Furthermore, we identify triplication of the kainate receptor (KAR) encoding gene Grik1 as the cause of these phenotypes. Normalization of Grik1 dosage in Ts2Cje mice specifically restored spatial memory and reversed the bidirectional alterations to CA1 inhibition, but not the changes in synaptic plasticity or the other behavioral modifications observed. We propose that modified information gating caused by disturbed inhibitory tone rather than generalized overinhibition underlies some of the characteristic cognitive deficits in DS.

[1] Instituto de Neurociencias CSIC-UMH, 03550 San Juan de Alicante, Spain. [2] Present address: LNC Therapeutics, 33000 Bordeaux, France.
*email: jlerma@umh.es

Down syndrome (DS) is caused by trisomy of human chromosome 21 (HSA21) and it is one of the most frequent genetic causes of intellectual disability[1]. Although the chromosomal origin of this syndrome was first described over half a century ago[2], we still do not really know what mechanisms link the triplication of the genes located on HSA21 to the cognitive deficits that characterize DS. These deficits include altered working memory, language and reading incompetence, impaired long-term memory, and spatial skills[3]. In recent decades, studies on human patients and mouse models have revealed a plethora of anatomical, cellular, and physiological alterations that might influence cognitive performance in DS. However, our understanding of the mechanisms underlying these deficits is still quite limited. Therefore, establishing a direct causal link between genetic, physiological, and intellectual alterations is an important hurdle that must be overcome to develop effective treatments for this syndrome. In this context, assessing the specific contribution of triplicated genes to specific brain alterations in DS is crucial to disentangle its multifaceted nature.

Among the physiological phenotypes characterizing DS, alterations to the excitatory–inhibitory balance are considered particularly important in the induction of associated cognitive deficits. Mouse models of DS present reduced excitatory drive[4,5] and, importantly, enhanced GABAergic activity in different brain regions, especially in the hippocampal formation[6–8]. This unbalanced excitatory and GABAergic activity has been invoked to explain alterations to synaptic plasticity and behavior, particularly as pharmacological blockade of GABA receptors somehow restores abnormal physiological and cognitive phenotypes in mouse models of DS[9–12]. Thus the so-called *GABAergic hypothesis for cognitive disabilities in DS* suggests that alterations in GABAergic systems increase inhibitory tone, underlying deficits in synaptic plasticity and cognitive functions[13,14]. However, a detailed explanation of the mechanisms by which excessive inhibition produces cognitive deficits has not yet been provided. Furthermore, we still lack a link between the triplication of specific gene(s) and the increased inhibitory tone in mouse models of DS.

The glutamate receptor gene *GRIK1* encodes the GluK1 subunit of KARs and it is triplicated in patients and mouse models of DS, as it is located on HSA21 and its murine ortholog, MMU16. In the hippocampus, GluK1 containing KARs are mainly expressed in GABA interneurons[15], where they can presynaptically regulate GABA release in a bidirectional manner[16–18], thereby modulating inhibitory control over hippocampal function in vivo[16]. *GRIK1*/GluK1 has also been shown to contribute to the physiopathology of epilepsy[19], a disorder characterized by an imbalance between excitation and inhibition, and a comorbidity of DS[20]. Therefore, *GRIK1* stands out as a good candidate to drive synaptic alterations in DS.

Here we assess how *Grik1* triplication affects the cognitive and behavioral deficits, synaptic plasticity, and basal synaptic transmission in the Ts2Cje model of DS[21]. By genetically normalizing the *Grik1* dosage in Ts2Cje mice, leaving the rest of the genes in the extra segment triplicated, we find that the additional dose of GluK1 is the cause for the spatial memory deficits evident in this model. Interestingly, these deficits are associated with a GluK1-dependent rearrangement in the somatodendritic inhibition of CA1 pyramidal cells but not to alterations in synaptic plasticity. By contrast, anxiety- and fear-related behavioral deficits are independent of *Grik1* triplication and they are not correlated to phenotypes of basal synaptic inhibition in the basolateral region of the amygdala (BLA). Overall, these data provide new clues as to how inhibition is spatially remodeled in DS, defining the role of this phenomenon in specific cognitive impairments.

## Results

**Grik1 triplication alters spatial memory and CA1 inhibition.** We first confirmed that *Grik1* is indeed overexpressed in Ts2Cje mice at both the mRNA and protein levels. As expected, more *Grik1* mRNA was detected in Ts2Cje trisomic animals than in their diploid littermates (Supplementary Fig. 1a). Moreover, an agonist of GluK1-containing KARs, ATPA, evoked larger currents when applied to CA1 interneurons of Ts2Cje mice, indicating an overexpression of KARs in the membrane of these cells (Supplementary Fig. 1b). ATPA did not evoke any current in pyramidal cells from CA1 or CA3 nor dentate gyrus (DG) granule cells, indicating the absence of misexpression in this mouse model (Supplementary Fig. 1c–e).

To assess the involvement of *Grik1* in the cognitive and synaptic abnormalities evident in the Ts2Cje model of DS, we used a genetic dose normalization strategy[7,22–26]. By breeding trisomic Ts2Cje females with disomic *Grik1*[+/−] heterozygous males, we specifically normalized *Grik1* dosage to euploid levels in a Ts2Cje trisomic background (Fig. 1a). The offspring consisted of disomic mice carrying two *Grik1* alleles (named euploid, which were used as controls), disomic mice heterozygous for *Grik1* (named Eu*Grik1*[+/−], not used in the experiments), trisomic mice carrying three *Grik1* alleles (named Ts2Cje), and trisomic mice with only two functional *Grik1* alleles and that therefore had a normalized dose of *Grik1* (named Ts*Grik1*[+/−]). We confirmed the normalization of *Grik1* dosage in Ts*Grik1*[+/−] mice by assessing the *Grik1* mRNA levels in the hippocampus (Fig. 1b). Ts2Cje mice did not display gross anatomical alterations, although they were lighter on postnatal days (P) 19–21 (Supplementary Fig. 2a). We did not observe any alterations to intrinsic cell parameters, such as the resting membrane potential, input resistance, or cell capacitance (Supplementary Fig. 2b). This mouse model did not show alterations in sensitivity gating or motor function (Supplementary Fig. 3). Furthermore, we did not detect a higher density of parvalbumin (PV[+]) or somatostatin (SOM[+]) interneurons in the hippocampus, amygdala or neocortex of Ts2Cje mice (Supplementary Figs. 4 and 5), in contrast to previous reports using different mouse models of DS[7].

We evaluated the memory of euploid, Ts2Cje, and Ts*Grik1*[+/−] animals by assessing their performance in the novel object recognition (NOR) and novel object location (NOL) paradigms[27]. Upon training with two objects, mice preferentially explore an object if it is a novel one or if it has been moved to a different location, yet only if they can remember the identity or location of the first object. In the NOL test, mice were trained in a square arena with two identical objects, exploring each of them equally (Supplementary Fig. 6a). One of the objects was then moved to a novel location in order to assess the animal's short-term spatial memory 1 h after training (Fig. 1c). While euploid mice preferentially explored the object in the novel location, the Ts2Cje mice failed to discriminate between the familiar and novel locations. However, normalization of the *Grik1* dosage was sufficient to restore spatial memory performance, as the behavior of the Ts*Grik1*[+/−] animals was identical to that of their euploid littermates. We also evaluated long-term spatial memory by assessing the response upon modification of the object location 24 h after training. Again, whereas euploid and Ts*Grik1*[+/−] mice both discriminated between the novel and familiar locations, Ts2Cje mice explored both objects equally (Fig. 1d). Therefore, triplication of *Grik1* impaired short- and long-term spatial memory performance in Ts2Cje animals, yet normalization of its gene dosage in Ts*Grik1*[+/−] mice reverted these phenotypes.

We next assessed mice performance in the NOR task. Here the mice were trained with two identical objects in a square arena, which they explored equally (Supplementary Fig. 6b). One of the

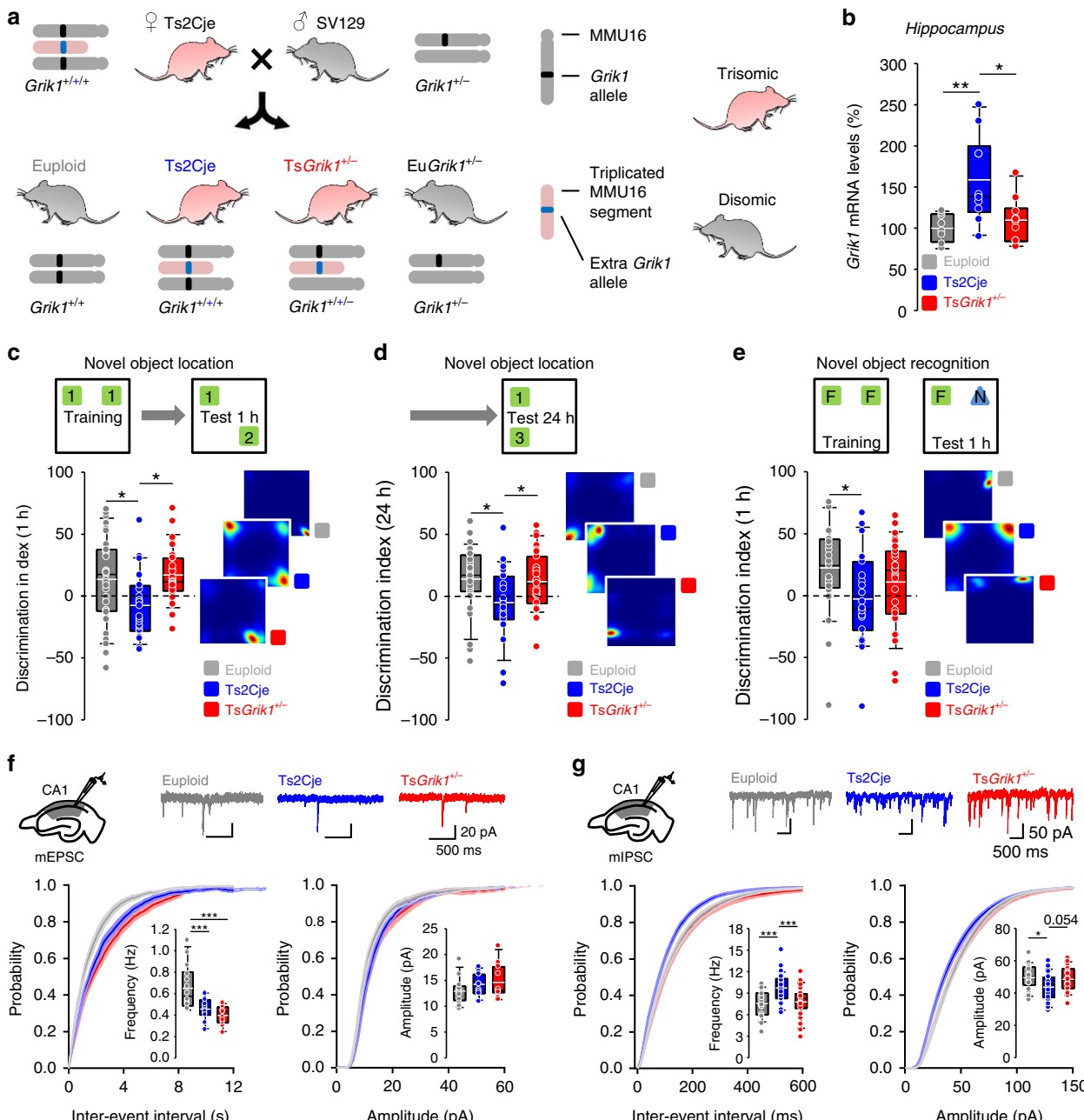

objects was then substituted by a novel one 1 h after training to assess short-term memory performance. As expected, the ability of Ts2Cje mice to discriminate between novel and familiar objects was impaired. In contrast to previous reports, showing a recovery of NOR performance upon genetic normalization of *Dyrk1a* and *Kcnj6* genes[25,28], restoration of the *Grik1* dosage in Ts*Grik1*[+/−] littermates did not fully recover this impairment to the levels of euploid mice (Fig. 1e). These results highlight the specific role of *Grik1* triplication in spatial memory deficits of Ts2Cje mice, a task that is directly related to hippocampal function.

To explore the mechanisms underlying the *Grik1*-dependent spatial memory deficits in Ts2Cje mice, we recorded synaptic activity in the CA1 region of hippocampal slices from euploid, Ts2Cje, and Ts*Grik1*[+/−] animals. When miniature excitatory postsynaptic currents (mEPSCs) were measured in CA1 pyramidal neurons, the frequency of the mEPSCs in Ts2Cje mice was lower than in their euploid littermates and *Grik1* normalization

did not restore this phenotype (Fig. 1f). This reduction in mEPSC frequency was not paralleled by spontaneous EPSCs (sEPSCs; Supplementary Fig. 7a) and the amplitude of m/sEPSCs was similar in each of the genotypes. When synaptic inhibition over CA1 pyramidal cells was assessed, an increase in the frequency and a decrease in the mean amplitude of miniature inhibitory postsynaptic currents (mIPSCs) was detected in Ts2Cje mice. Interestingly, these alterations reverted fully to the euploid levels in the Ts*Grik1*[+/−] mice (Fig. 1g). Similar *Grik1*-dependent alterations were detected in spontaneous IPSCs (sIPSCs: Supplementary Fig. 7b). Interestingly, the basal excitatory input to DG granule or CA3 pyramidal cells was not altered in Ts2Cje animals (Supplementary Fig. 8a, c). On the other hand, as found in other DS models, basal inhibition was increased in the DG (Supplementary Fig. 8b) and reduced in CA3 pyramidal neurons (Supplementary Fig. 8d), but both alterations were independent of *Grik1*. In conjunction, these results highlight the association

**Fig. 1** *Grik1*-dependent spatial memory impairment and anomalous basal inhibition in CA1 of Ts2Cje mice. **a** Experimental approach for *Grik1* dose normalization. We crossed trisomic Ts2Cje females with *Grik1*[+/−] males, obtaining offsprings consisting of wild-type mice (*Grik1*[+/+], euploid), trisomic mice with all genes in the extra segment triplicated (Ts2Cje), trisomic mice with all genes triplicated except *Grik1* (Ts*Grik1*[+/−]), and disomic mice monozygous for *Grik1* (*Grik1*[+/−]). **b** Real-time qPCR quantification of hippocampal *Grik1* mRNA levels in euploid, Ts2Cje, and Ts*Grik1*[+/−] mice. Expression levels were normalized to those of euploid animals ($n = 8$ for euploid, $n = 10$ for Ts2Cje, $n = 10$ for Ts*Grik1*[+/−], one-way ANOVA+Holm–Sidak). **c–e** Performance in the novel object location (NOL) and novel object recognition (NOR) paradigms. A schematic representation of the paradigms appears at the top, whereas the bottom displays box plots showing the quantification of data for each test session, and heat plots tracking representative animals' behavior. Ts2Cje mice did not discriminate between familiar and novel locations at short-term (1 h after training, **c**) and long-term (24 h after training, **d**) intervals in the NOL task. Normalization of *Grik1* dosage in Ts*Grik1*[+/−] mice restored the memory performance to euploid levels ($n = 31$ for euploid, $n = 23$ for Ts2Cje, $n = 27$ for Ts*Grik1*[+/−], one-way ANOVA+Holm–Sidak). **e** Ts2Cje mice presented impaired discrimination between familiar and novel objects in the NOR test 1 h after training. *Grik1* dosage normalization partially rescued this impairment ($n = 29$ for euploid, $n = 23$ for Ts2Cje, $n = 27$ for Ts*Grik1*[+/−], one-way ANOVA+Holm–Sidak). **f** Miniature excitatory postsynaptic currents (mEPSCs) were recorded at −75 mV in hippocampal CA1 pyramidal neurons to evaluate basal excitatory synaptic transmission. The schematic representation of the electrode arrangement in the hippocampus and representative traces of mEPSC recordings are shown in the top. Bottom, cumulative distribution and box plots showing reduced mEPSC frequency (left) and unaltered amplitude (right) in Ts2Cje and Ts*Grik1*[+/−] mice ($n = 14$ for euploid, $n = 11$ for Ts2Cje, $n =$ for Ts*Grik1*[+/−], one-way ANOVA+Holm–Sidak). **g** Miniature inhibitory postsynaptic currents (mIPSCs) were recorded at −60 mV in CA1 pyramidal neurons to evaluate basal inhibitory transmission. Top, schematic representation of electrode arrangement and representative traces of mIPSCs. Bottom, cumulative probability and box plots showing increased frequency (left) and reduced amplitude (right) of mIPSCs in Ts2Cje mice. Normalization of *Grik1* dosage recovered these phenotypes to euploid levels ($n = 22$ for euploid, $n = 31$ for Ts2Cje, $n = 23$ for Ts*Grik1*[+/−], one-way ANOVA+Holm–Sidak). $*p < 0.05$, $**p < 0.01$, $***p < 0.005$. For detailed data values and statistics, see Supplementary Table 2

between *Grik1*-dependent spatial memory impairments and synaptic inhibition but not excitation specifically in the CA1 hippocampal area of Ts2Cje mice.

**Deficits in long-term potentiation (LTP) are independent of the degree of inhibition.** Reduced LTP is the most prominent alteration to synaptic plasticity reported in mouse models of DS[9,29,30], and it is known that pharmacological blockade of inhibitory activity restores LTP deficits, suggesting that they are caused by excessive GABAergic inhibition[10,11]. Our genetic normalization strategy identified *Grik1* to be the cause of the inhibitory phenotypes found in the CA1 region of Ts2Cje mice, allowing us to directly address this suggestion. LTP and depotentiation in the CA1 of Ts2Cje mice were assessed by measuring the CA1 extracellular field excitatory postsynaptic potentials (fEPSPs) after stimulating Schaffer collaterals. Induction of LTP with a classic high-frequency stimulation (HFS) protocol produced weaker plasticity in Ts2Cje animals than in their euploid littermates. Surprisingly, this weaker plasticity persisted in Ts*Grik1*[+/−] mice, in which the *Grik1* dosage and basal inhibitory synaptic transmission were restored to euploid levels (Fig. 2a). Applying a depotentiation protocol after LTP induction to mimic synaptic depression in potentiated synapses produced a similar effect in each genotype (Fig. 2b).

The LTP deficits in mouse models of DS may depend on the stimulation protocol used[30]. However, inducing LTP with a more natural theta-burst stimulation (TBS) protocol again produced a reduced synaptic potentiation in Ts2Cje and Ts*Grik1*[+/−] mice (Fig. 2c). Likewise, depotentiation after TBS-induced LTP was similar in all genotypes (Fig. 2d). Accordingly, and unlike the blunter pharmacological blockade, more precise reversion of synaptic inhibition to euploid levels did not reverse the deficits in synaptic plasticity, suggesting that this phenomenon is independent of the degree of inhibition.

**Bidirectional modulation of GABA release in the CA1 region.** To further understand the alterations to basal inhibitory transmission in Ts2Cje mice and its possible impact in spatial memory, we undertook a detailed analysis of the properties of mIPSCs impinging onto CA1 pyramidal cells. Different inhibitory neurons contact these cells at specific regions along their somatodendritic axis, strongly modulating excitatory input and the firing

of action potentials[31]. In terms of amplitude and kinetics, the properties of inhibitory synaptic responses are constant at synaptic sites along the pyramidal cell dendrites in the CA1 region[32]. However, IPSCs that impinge on distal regions of CA1 pyramidal cells undergo dendritic filtering, which reduces their amplitude and slows their kinetics[33]. Thus dendritic filtering provides an opportunity to evaluate the origin of IPSCs according to their amplitude and kinetic properties. Therefore, we sorted mIPSCs into 20-pA bins, using the amplitude of these events as an estimate of their origin in the somatodendritic axis. We found that there was an increase in the relative frequency of events <20 pA in Ts2Cje mice, with a lower proportion of mIPSCs with amplitudes between 40 and 80 pA. These alterations were caused by *Grik1* triplication as they were reversed in Ts*Grik1*[+/−] littermates (Fig. 3a). Similarly, binning the events into small, medium, and large categories (0–20, 20–40 and 40–80 pA, respectively) yielded equivalent results (Fig. 3b). The bidirectional modulation of the relative frequencies of mIPSCs in Ts2Cje mice suggests that *Grik1* triplication affects GABA release distinctly along the somatodendritic axis of CA1 pyramidal cells, although we cannot discard that altered synapse density might also contribute to alterations in mIPSC frequency and amplitude.

Besides amplitude, dendritic filtering also affects the kinetics of distally originated synaptic responses. Indeed, the rise and decay times of mIPSCs impinging onto distal regions of CA1 pyramidal cells are slower than those contacting proximal and somatic areas[33]. Thus inhibitory synaptic responses were further analyzed according to their somatodendritic origin by sorting the mIPSCs according to their rise times into 1-ms time bins. There was a lower proportion of fast events and a higher proportion of slow events in Ts2Cje mice, and this phenotype reverted to euploid levels in the Ts*Grik1*[+/−] littermates (Fig. 3c).

To further demonstrate the GluK1-dependent bidirectional alterations in distal and proximal inhibitory synapses in the Ts2Cje CA1, the relative frequency of small and slow (hereafter distal) events in these mice were assessed relative to the large and fast (hereafter proximal) events. There was an increase in the proportion of distal events in Ts2Cje animals compared to their euploid littermates, a phenotype that was reversed in Ts*Grik1*[+/−] mice (Supplementary Fig. 9a). By contrast, the proportion of proximal events was diminished in Ts2Cje animals relative to their euploid and Ts*Grik1*[+/−] littermates (Supplementary Fig. 9b). Hence, in contrast to the generally accepted view in mouse

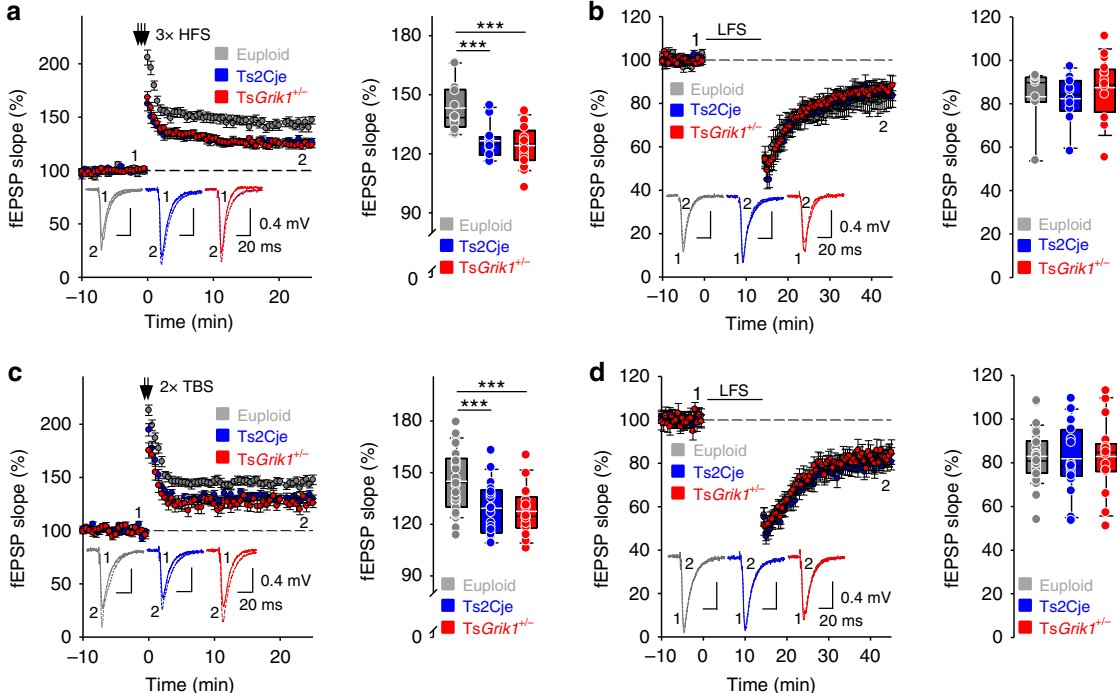

**Fig. 2** Reduced LTP in Ts2Cje mice is independent of *Grik1*. **a** LTP was induced by high frequency stimulation (HFS; 3 × 100 Hz, 1 s) in euploid, Ts2Cje and Ts*Grik1*$^{+/−}$ mice ($n = 13$ for euploid, $n = 11$ for Ts2Cje, $n = 20$ for Ts*Grik1*$^{+/−}$, one-way ANOVA+Holm–Sidak). In this and the following panels, insets show representative examples of field excitatory postsynaptic potentials (fEPSPs) in each situation. Ts2Cje and Ts*Grik1*$^{+/−}$ mice presented reduced HFS-induced LTP than euploid littermates. **b** Unaltered depotentiation after HFS-induced LTP in transgenic mice ($n = 7$ for euploid, $n = 10$ for Ts2Cje, $n = 16$ for Ts*Grik1*$^{+/−}$, one-way ANOVA). **c** LTP was induced by theta-burst stimulation (TBS, 2 × 4 stimuli at 100 Hz, repeated 10 times at 5 Hz) in euploid, Ts2Cje, and Ts*Grik1*$^{+/−}$ mice. Again, Ts2Cje and Ts*Grik1*$^{+/−}$ mice showed reduced LTP over euploid littermates ($n = 32$ for euploid, $n = 20$ for Ts2Cje, $n = 17$ for Ts*Grik1*$^{+/−}$, one-way ANOVA+Holm–Sidak). **d** Depotentiation was induced by applying 900 stimuli at 1 Hz after TBS-induced LTP and there were no differences between genotypes ($n = 31$ for euploid, $n = 20$ for Ts2Cje, $n = 17$ for Ts*Grik1*$^{+/−}$, one-way ANOVA). For detailed data values and statistics, see Supplementary Table 4. ***$p < 0.005$

models of DS, excessive synaptic inhibition is not ubiquitous but rather it appears to be segregated to distal dendrites in the CA1 region while proximal somatodendritic segments become disinhibited.

To address this issue in more detail, we assessed the total charge of the inhibitory current associated with the distal and proximal events in each genotype. As expected, we found a remarkable increase (226%) in charge transfer due to distal mIPSCs in Ts2Cje mice, a phenotype that was restored to euploid levels in Ts*Grik1*$^{+/−}$ littermates (Fig. 2d, left panel). By contrast, the total charge transfer conveyed by proximal mIPSCs was reduced by 20% in Ts2Cje mice relative to the euploid and Ts*Grik1*$^{+/−}$ animals (Fig. 3d, right panel), further supporting the existence of GluK1-dependent bidirectional alterations to GABA release in the CA1 region of Ts2Cje mice. We found similar bidirectional alterations to GABA release in relation to proximal and distal inhibitory synapses upon in-depth analysis of sIPSCs (Supplementary Fig. 10).

**Bidirectionally modified GABA release onto CA1 neurons.** In light of the above, we assessed the GABA release probability over the somatodendritic axis of CA1 pyramidal neurons, measuring the paired pulse ratio (PPR) of evoked IPSCs over the proximal and distal dendritic regions of these cells. Stimulation of distal inhibitory terminals revealed a decreased PPR in Ts2Cje animals, indicating a higher probability of GABA release in this area (Fig. 4a), and this parameter was restored to euploid levels in Ts*Grik1*$^{+/−}$ mice. Conversely, proximal stimulation showed increased PPR in Ts2Cje mice relative to the levels of euploid mice and their Ts*Grik1*$^{+/−}$ littermates, indicating a lower GABA

release probability (Fig. 4b). These results corroborate the analysis of the m/sIPSC data, providing compelling evidence that *Grik1* triplication provokes an imbalance in synaptic inhibition over different dendritic segments of CA1 pyramidal neurons in Ts2Cje mice. This imbalance in inhibition had been previously overlooked and it may provoke deficits in information processing in this region[34] that could in turn underlie the impaired spatial memory observed in mouse models of DS.

**Reduced anxiety/fear is independent of *Grik1* triplication.** The involvement of *Grik1* in hyperactivity, anxiety, and fear, three core alterations in mouse models of DS[35], was assessed using a battery of behavioral tests. The behavior of euploid, Ts2Cje, and Ts*Grik1*$^{+/−}$ mice was first analyzed in the open field arena, and Ts2Cje and Ts*Grik1*$^{+/−}$ mice were seen to travel a greater distance than their euploid littermates (Fig. 5a, left panel), excluding a role of *Grik1* in the hyperactivity described in mouse models of DS[10]. Ts2Cje and Ts*Grik1*$^{+/−}$ mice also spent longer in the center of the open field, a parameter related to anxiety-like behaviors and suggesting less anxiety in these animals (Fig. 5a, right panel). Anxiety-related phenotypes were further explored in an elevated plus maze, and consistent with previous reports in other models of DS[35,36], Ts2Cje mice spent more time in the open arms than their euploid littermates. This phenotype was independent of *Grik1* as it was not reversed in Ts*Grik1*$^{+/−}$ animals (Fig. 5b). In this test, the Ts2Cje and Ts*Grik1*$^{+/−}$ mice traveled a greater distance than their euploid littermates (Supplementary Fig. 11a).

Finally, contextual and cued fear conditioning was assessed in Ts2Cje and Ts*Grik1*$^{+/−}$ mice. During a training session, mice in a conditioning cage were presented with an auditory tone

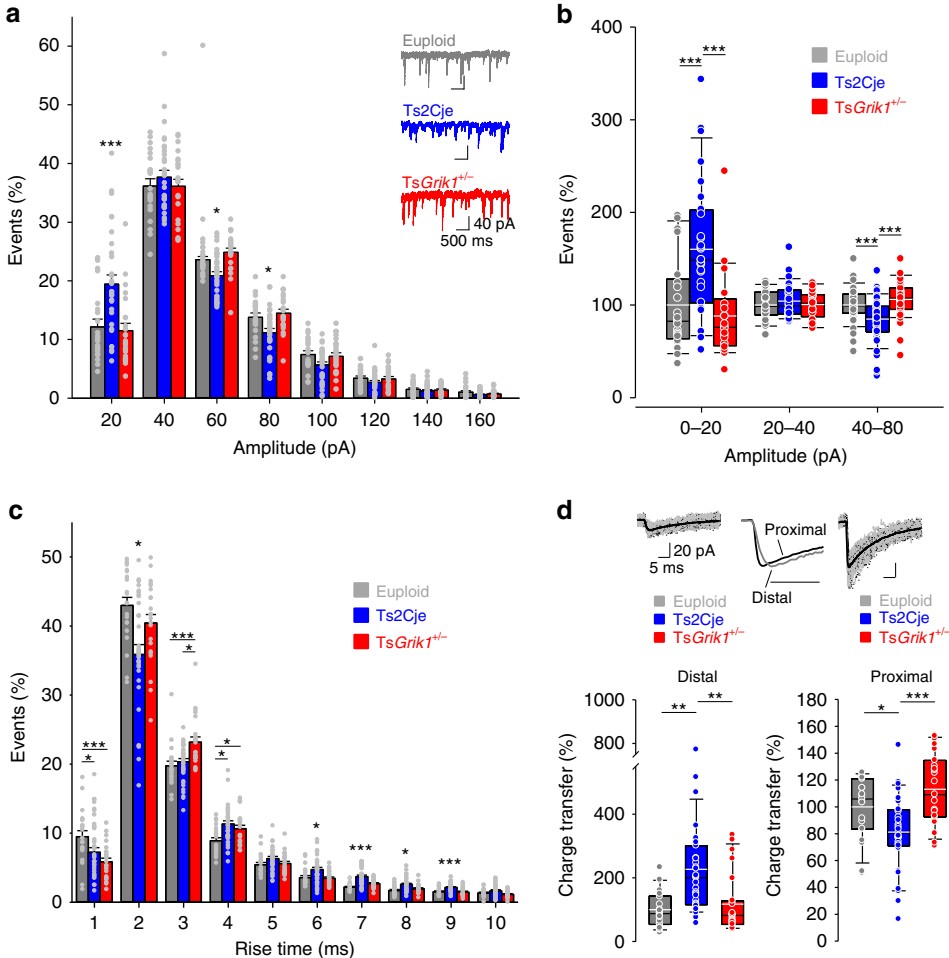

**Fig. 3 Bidirectional alterations of inhibitory synaptic transmission in CA1. a** Frequency histogram for mIPSCs recorded in CA1 pyramidal cells sorted in 20 pA bins to assess the relative proportion of small and large events across genotypes. The insets show representative recordings of mIPSC from the three different genotypes. CA1 pyramidal neurons of Ts2Cje mice presented reduced and increased proportion of small (<20 pA) and large (40–80 pA) mIPSCs, respectively, than observed in euploid and TsGrik1[+/−] littermates (n = 21 for euploid, n = 31 for Ts2Cje, n = 23 for TsGrik1[+/−], one-way ANOVA +Holm–Sidak). **b** Sorting of CA1 mIPSC events in small, medium-sized, and large categories. Box plots illustrate increased contribution of small events and decreased relative frequency of large events in Ts2Cje mice compared to euploid and TsGrik1[+/−] littermates (n = 21 for euploid, n = 31 for Ts2Cje, n = 23 for TsGrik1[+/−], one-way ANOVA+Holm–Sidak). **c** Frequency histogram of mIPSCs in CA1 sorted in 1 ms rise time bins to assess the relative proportion of fast and slow events across genotypes. The CA1 pyramidal neurons of Ts2Cje mice received reduced and increased proportion of fast and slow mIPSCs, respectively, than euploid and TsGrik1[+/−] littermates (n = 21 for euploid, n = 31 for Ts2Cje, n = 23 for TsGrik1[+−], one-way ANOVA+Holm–Sidak). **d** Top insets are examples of distal (small and slower) and proximal (large and faster) mIPSCs. The middle recording shows averaged mIPSC superimposed after normalization. At the bottom, the box plots show the charge transfer carried by small and slow inhibitory synaptic events (left) and by large and fast mIPSCs (right). In both cases, data were normalized to euploid values and presented as percentage (n = 22 for euploid, n = 31 for Ts2Cje, n = 23 for TsGrik1[+/−], ANOVA on ranks+Dunn's). In bar plots, error bars represent the s.e.m. For detailed data values and statistics, see Supplementary Table 3. *p < 0.05, **p < 0.01, ***p < 0.005

(conditioned stimulus (CS)) immediately before a foot shock (unconditioned stimulus (US)). In the training session, all the mice froze in response to the US, irrespective of the genotype (Supplementary Fig. 11b). In the contextual fear test carried out 24 h after training, Ts2Cje and TsGrik1[+/−] mice showed milder freezing responses (Supplementary Fig. 11c), suggesting impaired contextual memory formation or association with the US. The mice were later placed in a different context where they were presented with the CS, and again the Ts2Cje and TsGrik1[+/−] mice froze less in response to the auditory tone (Supplementary Fig. 11d), ruling out an influence of Grik1 on fear-related responses in these animals. However, taking advantage that the same mice were subjected to different behavioral tests, we noticed a correlation between contextual fear conditioning and NOL performance in euploid but not in

trisomic animals, indicating that contextual fear conditioning relies on spatial memory only in the euploid condition (Supplementary Fig. 12a, b). By contrast, a negative correlation between anxiety-related phenotypes (time in the open arms of the elevated plus maze) and contextual fear conditioning responses was evident in trisomic but not in euploid mice (Supplementary Fig. 12c, d). A similar negative correlation was found between anxiety-related phenotypes and cued fear conditioning in trisomic but not in euploid mice (Supplementary Fig. 12e, f). Together, although not definitive, these results suggest that reduced anxiety may influence fear memory trials in trisomic mice, hampering interpretations of fear conditioning tests in these animals. We believe that some caution should be taken at the time of interpreting fear memory and related behaviors in a context of altered anxiety.

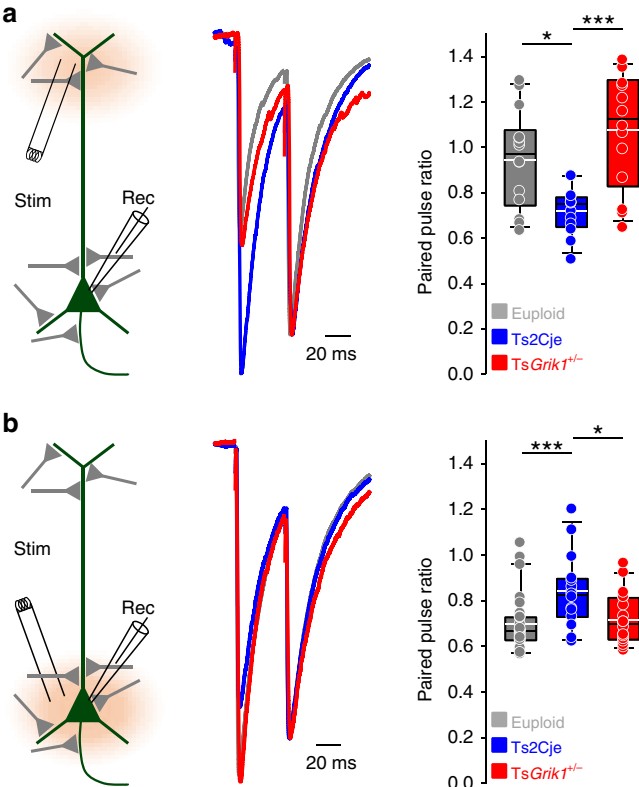

**Fig. 4** The probability of GABA released onto CA1 pyramidal cells is altered in Ts2Cje mice. **a** Evoked IPSCs (eIPSCs) by inhibitory terminals at distal regions of the CA1 pyramidal cell dendrites. Left, layout of the positions of recording and stimulation electrodes. Middle, sample traces normalized to the second response and superimposed, from the three different genotypes. Right, box plot showing that the paired pulse ratio was reduced in Ts2Cje mice—indicating increased probability of GABA release—and its recovery upon *Grik1* normalization ($n = 14$ for euploid, $n = 13$ for Ts2Cje, $n = 14$ for Ts*Grik1*$^{+/-}$, one-way ANOVA+Holm-Sidak). **b** Paired pulse ratio of eIPSCs obtained by stimulating inhibitory terminals proximal to the CA1 pyramidal cell soma. Left, position of stimulation and recording electrodes. Middle, sample traces normalized to the second response. Right, box plot showing increased paired pulse ratio—indicating reduced probability of GABA release—in Ts2Cje and its recovery to euploid levels in Ts*Grik1*$^{+/-}$ littermates ($n = 27$ for euploid, $n = 15$ for Ts2Cje, $n = 19$ for Ts*Grik1*$^{+/-}$, one-way ANOVA+Holm-Sidak). For detailed data values and statistics, see Supplementary Table 3. $*p < 0.05$, $***p < 0.005$

Despite the deficits in anxiety and fear described in mouse models of DS, the possible alterations to basal synaptic transmission in the amygdala that might underlie such effects have not been explored previously, this region having classically been associated with anxiety and fear[37]. To assess the possible synaptic correlates of impaired anxiety and fear, we evaluated basal excitation and inhibition in the BLA of euploid, Ts2Cje, and Ts*Grik1*$^{+/-}$ mice. GluK1 is present at excitatory and inhibitory terminals in this region[38,39], providing the potential to modulate the excitability of the amygdala complex and to thereby regulate anxiety and fear responses. Unlike the hippocampus, neither the frequency nor the amplitude of mEPSCs changed in pyramidal neurons from any of the mice studied (Fig. 5c). Furthermore, the frequency and amplitude of mIPSCs was similar in each genotype (Fig. 5d). Finally, when the contribution of putative distal and proximal mIPSCs to basal inhibition of pyramidal neurons in the BLA was assessed there was a similar relative frequency of distal and proximal events in each genotype (Fig. 5e). Accordingly, we found unaltered charge transfer over distal and proximal dendritic segments across genotypes (Fig. 5f). These data further highlight that the anxiety and fear phenotypes in the Ts2Cje mouse model are independent of *Grik1*, indicating that basal synaptic activity does not influence the phenotype of these mice.

## Discussion

In this study, we describe an alteration in GABA release over proximal and distal regions of the CA1 pyramidal cell somato-dendritic axis in the Ts2Cje mouse model of DS. This alteration was bidirectional, with increased and decreased GABA release over distal and proximal dendrites, respectively. This observation widens our understanding of GABAergic abnormalities in DS and forces us to reconsider the generally accepted view of a simple overinhibition of principal cells in DS[13,14]. Rather, our data demonstrate that trisomy results in a new regime of dendritic inhibition that presumably drives distorted dendritic integration of excitatory inputs, affecting hippocampal performance[34,40,41] and probably underlying the spatial memory deficits in this mouse model. We hypothesize that this new regime of GABA release onto separated dendritic segments segregates incoming excitatory information from different sources in the trisomic animals. Thus it may favor associational over temporoammonic inputs in the hippocampal CA1, since proximal inhibition is depressed while apical inhibition, which is mainly carried by oriens-lacunosum moleculare (OLM) interneurons, is largely potentiated. Indeed, KARs control OLM activity[42] and it was recently shown that optogenetic activation of these cells affects information gating in the CA1, causing memory impairment[43,44]. We speculate that *Grik1* overexpression mimics OLM interneuron optogenetic activation, causing the spatial memory impairment observed in Ts2Cje mice. This situation may be equivalent to the change in spatial references induced by alternating basal and apical dendritic inhibition of pyramidal neurons in the entorhinal cortex[34]. We further identify *Grik1* triplication to be responsible for the altered basal inhibitory synaptic activity in the CA1 region of these mice. In the hippocampus, GluK1 is present in GABAergic terminals[15,17] where it may bidirectionally regulate GABA release through its canonical and non-canonical signaling pathways[16,17,45].

We used a dosage normalization strategy to assess the role of the KAR subunit-encoding gene *Grik1* in the cognitive, behavioral, and physiological phenotypes displayed by the Ts2Cje model of DS. By breeding Ts2Cje females with disomic *Grik1*$^{+/-}$ males, we were able to normalize the dosage of *Grik1* on a trisomic Ts2Cje background, which allowed us to directly interrogate about the impact of altered inhibition in cognition. This normalization of *Grik1* dosage restored spatial memory deficits and reverted the alterations to basal inhibitory transmission in the CA1 area of the hippocampus, two phenotypic features of murine models of DS. Unexpectedly, these changes were not associated with the recovery of deficits in synaptic plasticity, arguing against the widely accepted connection between hippocampal synaptic inhibition and LTP in mouse models of DS[10,11]. Our data also indicates that synaptic inhibition and spatial memory are independent from the LTP magnitude in the CA1 region of this model of DS. This lack of a correlation between LTP and spatial memory has been indicated previously[46,47], and questions LTP to be used as a correlate for spatial memory in mouse models of DS. Indeed, we found LTP to be reduced but not abolished in trisomic animals, meaning that such a reduced LTP does not prevent spatial learning. Nevertheless, the lack of a correlation between the degree of inhibitory synaptic transmission and LTP suggests a different causal mechanism for the latter

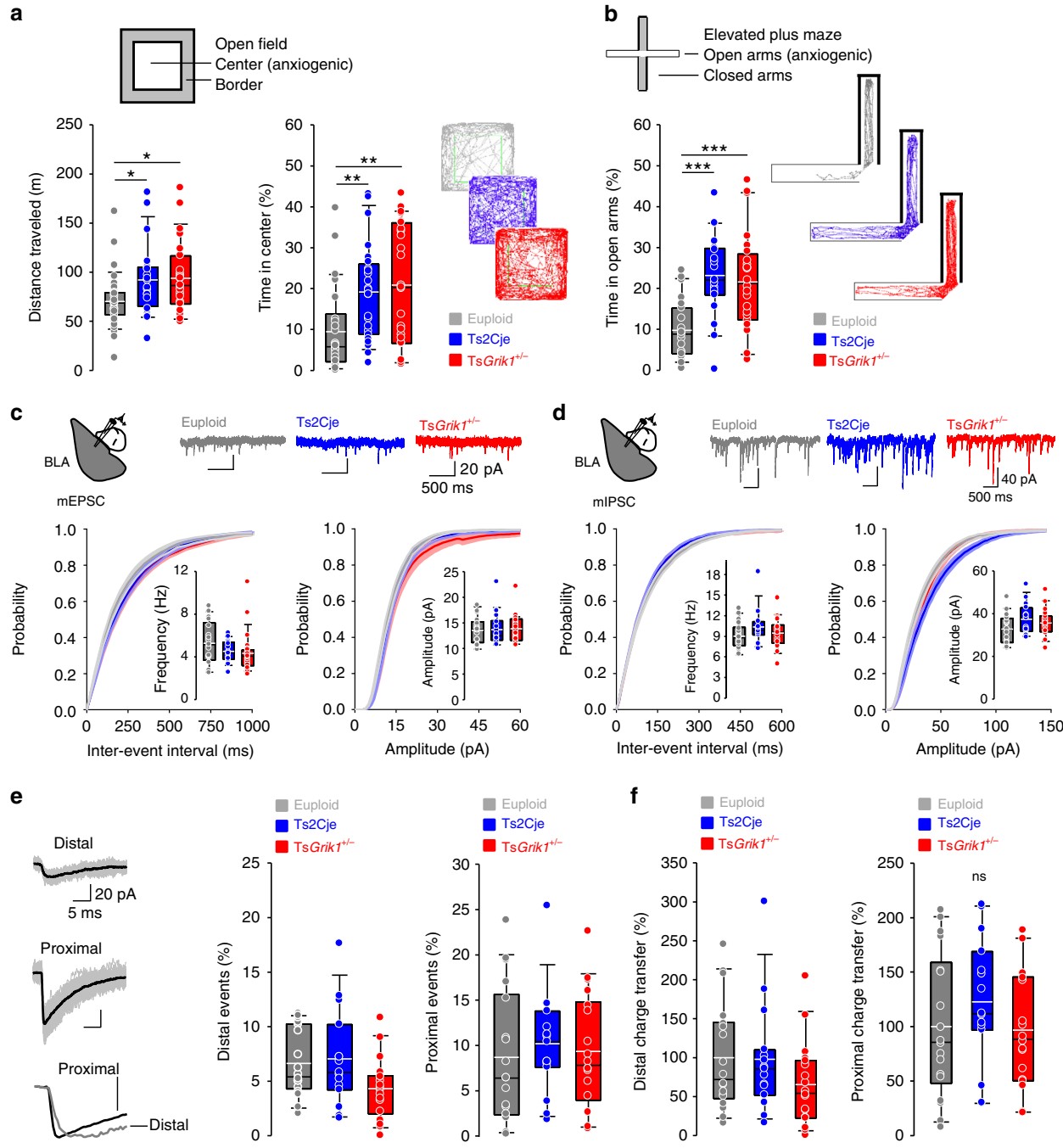

in this model, one of them could be the reduced glutamatergic transmission in the DS syndrome model.

Trisomic animals present impaired short- and long-term spatial and recognition memory, as well as poor performance in executive tasks associated with working memory[35]. Our normalization strategy identified *Grik1* triplication not only as the cause of redistribution of inhibitory power in the CA1 pyramidal cells but also as the cause for the spatial memory deficits in Ts2Cje mice. While Ts2Cje animals did not remember previous object locations in the NOL paradigm, normalization of *Grik1* dosage reversed this phenotype and these mice performed like the euploid mice following both short- and long-term delays. By contrast, *Grik1* normalization did not fully restore the performance of Ts2Cje mice in the NOR test. This result may be explained by the fact that

triplication of *Kcnj6*, a voltage-gated K$^+$ channel, seems to be also a major cause for NOR deficits in Ts65Dn mice[25].

There are several reports indicating that enhanced basal GABAergic transmission in the hippocampal DG[6] and CA1[7,48] is a core feature in mouse models of DS[13,14]. A higher density of inhibitory terminals[49] and/or GABAergic neurons[7,49] has been proposed to explain these phenotypes, yet the number of inter-neurons in human DS patients are unaltered or slightly reduced[50,51]. Similarly, no increase was evident in the density of SOM$^+$ or PV$^+$ interneurons in CA1, CA3, DG, or neocortex of Ts2Cje mice, suggesting that the Ts2Cje model may be closer to the human condition with regard to this specific phenotype than the gold-standard Ts65Dn. Although the levels of expression of PV may be activity dependent, making cell counting uncertain,

**Fig. 5** Anxiety and fear phenotypes are independent of *Grik1* triplication in Ts2Cje mice. **a** Top, schematic representation of the open field arena and the areas considered border (shaded) and center (light). Bottom, box plot showing quantification of distance traveled (left) and time in center (right) as well as representative mice tracks in the arena. Ts2Cje and Ts*Grik1*[+/−] mice showed increased distance traveled and time in center of the arena, indicating hyperlocomotion and reduced anxiety ($n = 35$ for euploid, $n = 24$ for Ts2Cje, $n = 27$ for Ts*Grik1*[+/−], ANOVA on ranks+Dunn's). **b** Top, schematic representation of the elevated plus maze, with two open (light) and two closed (shaded) arms. Bottom, box plot showing quantification of time spent in open arms and representative mice tracks in the maze. Ts2Cje and Ts*Grik1*[+/−] mice spent increased time in the open arms, indicating reduced anxiety ($n = 23$ for euploid, $n = 19$ for Ts2Cje, $n = 24$ for Ts*Grik1*[+/−], one-way ANOVA+Holm–Sidak). **c** Basal excitatory synaptic transmission (mEPSCs) recorded at −75 mV in basolateral amygdala (BLA) neurons. Top, schematic representation of the area recorded and representative traces of mEPSCs. Bottom, cumulative distribution and box plots showing unaltered mEPSC frequency (left) and amplitude (right) in Ts2Cje and Ts*Grik1*[+/−] mice ($n = 21$ for euploid, $n = 18$ for Ts2Cje, $n = 24$ for Ts*Grik1*[+/−], ANOVA on ranks). **d** Basal inhibitory transmission (mIPSCs) recorded at −60 mV in BLA neurons. Top, schematic representation of recorded area and representative traces of mIPSCs. Bottom, cumulative probability and box plots showing unaltered frequency (left) and amplitude (right) of mIPSCs recorded in Ts2Cje mice ($n = 18$ for euploid, $n = 15$ for Ts2Cje, $n = 18$ for Ts*Grik1*[+/−]). **e** Left, representative traces of distal (top), proximal (middle), and normalized (bottom) mIPSCs recorded in BLA neurons. Box plots showing quantification of proportion of distal (center) and proximal (right) mIPSCs ($n = 18$ for euploid, $n = 15$ for Ts2Cje, $n = 18$ for Ts*Grik1*[+/−], one-way ANOVA in distal mIPSCs, ANOVA on ranks in proximal mIPSCs). **f** Box plots representing the charge carried by distal (left) and proximal (right) mIPSCs in BLA neurons ($n = 18$ for euploid, $n = 15$ for Ts2Cje, $n = 18$ for Ts*Grik1*[+/−], ANOVA on ranks in distal mIPSCs, one-way ANOVA in proximal mIPSCs). For detailed data values and statistics, see Supplementary Table 5. *$p < 0.05$, **$p < 0.01$, ***$p < 0.005$

our results do not support the prevailing idea of increased interneuron density in DS mouse models.

Another key finding here is that *Grik1*-dependent phenotypes appear in a region-specific manner, circumscribed to the hippocampal CA1. Previous reports showed deficits related to locomotor activity, anxiety, and fear in different mouse models of DS[35]. Locomotion of Ts2Cje mice was enhanced in the open field test, with less anxiety and fear in the elevated plus maze and fear conditioning tests. None of these phenotypes were restored upon normalization of the *Grik1* dosage and we could not find any correlate for these alterations at the level of basal synaptic activity in the amygdala. These results highlight the specific role of *Grik1* in hippocampal- but not amygdala-dependent phenotypes. We believe that the laminar organization of the hippocampal formation[31] constitutes a substrate for layer-specific alterations, such as those described in the CA1 of Ts2Cje mice. In this context, KAR-mediated bidirectional alterations may play an important role in CA1 physiology and consequently on its function. These bidirectional alterations may be absent or play negligible roles in determining behavioral phenotypes in regions without a similar laminar organization, such as the BLA.

In recent decades, alterations to inhibitory systems have been highlighted as a core feature in the pathophysiology of DS. The so-called *GABAergic hypothesis for cognitive impairment in DS* was postulated on the back of results obtained following pharmacological blockade of different GABA receptors in mouse models, paving the way to test these blockers in human patients[13]. Unfortunately, these clinical trials have been unsuccessful and some were prematurely abandoned owing to a lack of effect in patients. We believe that this failure was due to the non-specific nature of pharmacological blockade of GABAergic activity. Our results highlight a previously unknown complexity in hippocampal synaptic inhibition, advocating specific strategies to address the physiological and cognitive implications of synaptic inhibition in mouse models of DS. We identified the triplication of a single gene, *Grik1*, as the cause of altered inhibitory transmission in the Ts2Cje mouse model of DS, defining its role in cognitive impairment. We firmly believe that assessing the role of particular genes in specific phenotypes (e.g., ref. [46]) is the key to understand and to eventually treat cognitive impairments in DS.

## Methods

**Mice**. All experiments were performed according to Spanish and European Union regulations regarding animal research (2010/63/EU), and the experimental procedures were approved by the Bioethical Committee at the Instituto de Neurociencias and the Consejo Superior de Investigaciones Científicas. Animals were housed in ventilated cages in a standard pathogen-free facility, with free access to

food and water, controlled temperature (23 °C) and humidity (40–60%), and on a 12 h light/dark cycle.

Ts2Cje (stock number 004850) females and B6EiC3SnF1/J males (stock number 001875) were acquired from the Jackson Laboratory (Maine, USA) and bred to expand and maintain the Ts2Cje female colony. The resulting Ts2Cje females were crossbred with *Grik1* heterozygous males[52] (maintained on a SV129 background) to obtain the experimental animals. The mice were unambiguously tagged at the moment of weaning (P15) and ear punch residues were used for genotyping.

**Antibodies, chemicals, and other reagents**. The primary and secondary antibodies used here were anti-parvalbumin (polyclonal, 1:1000, rabbit, Abcam, Cat#ab11427), anti-somatostatin (polyclonal, 1:500, rabbit, ImmunoStar, Cat#20067), and Alexa Fluor 488-labeled anti-rabbit 1:1000, goat, Molecular Probes, Cat#A-11034). The reagents and drugs used were picrotoxin (Sigma, Cat#P1675), tetrodotoxin (TTX; Abcam, Cat#ab120055), LY303070 (ABX, Cat#5500.1500), and D-APV (Abcam, Cat#ab120003).

**Slice electrophysiology**. Brain slices were obtained as indicated elsewhere[37,53]. P19–21 animals were used for patch-clamp experiments, whereas animals at ~P60 were used for synaptic plasticity experiments. Briefly, the brain of the mice was extracted in ice-cold sucrose artificial cerebrospinal fluid (sucrose ACSF, containing, in mM: 248 sucrose, 26 $NaHCO_3$, 10 glucose, 3 KCl, 1.25 $NaH_2PO_4$, 4 $MgSO_4$, and 0.5 $CaCl_2$) and they were cut along the sagittal or coronal plane (CA1 and BLA experiments, respectively). The slices were transferred to a pre-warmed (32 °C) chamber with ACSF (containing, in mM: 124 NaCl, 26 $NaHCO_3$, 10 glucose, 3 KCl, 1.25 $NaH_2PO_4$, 1 $MgSO_4$, and 2 $CaCl_2$) and they were allowed to recover for 1 h prior to recording. For electrophysiological experiments, slices were transferred to a submersion chamber constantly perfused with carbogen-saturated ACSF and visualized using a differential interference contrast microscope (Leica DM LFSA). In hippocampal and BLA whole-cell experiments to assess inhibitory activity (the m/sIPSCs and PPR of evoked IPSCs), 3–5 MΩ resistance borosilicate glass pipettes were filled with internal solution containing (in mM: 130 CsCl, 8 NaCl, 10 HEPES, 5 QX314, 0.5 EGTA, and 4 ATP. APV (25 μM) and GYKI 53655 (25 μM) were added to the ACSF to block *N*-methyl-D-aspartate and AMPA receptors, respectively, and the membrane potential was maintained at −60 mV. When miniature synaptic potentials were measured, TTX (500 nM) was applied to block action potential firing. In hippocampal and BLA whole cell experiments to assess excitatory activity (m/sEPSCs), the pipettes were filled with internal solution containing (in mM: 130 CsMeSO₃, 4 NaCl, 10 HEPES, 10 TEA, 1 EGTA, 5 QX314, 2 ATP, and 0.5 GTP. Picrotoxin (50 μM) was applied to the ACSF to block $GABA_A$ receptors and the membrane potential was clamped at −75 mV. In hippocampal CA1 experiments to determine the firing properties of pyramidal cells, recording pipettes were filled with an internal solution containing (in mM: 135 $KMeSO_3$, 4 $MgCl_2$, 10 HEPES, 4 ATP, 0.4 GTP, and 5 phosphocreatine. In these experiments, cells were recorded in current clamp configuration. Immediately after whole-cell access, the resting membrane potential ($V_m$) was measured and then current steps from −50 to 300 pA were applied to the cells. The input resistance was calculated by injecting a −50-pA current step and dividing the resulting voltage drop by the amplitude of the current injected. Cell capacitance was calculated by inducing a 5-mV voltage step and dividing the resulting charge transfer by the amplitude of the voltage step. We considered CA1 interneurons to be those cells lying out of the pyramidal layer, mainly within the stratum oriens.

In experiments evoking IPSCs in the CA1 region, paired stimuli were delivered at 40-ms interval and at a frequency of 0.01 Hz. The stimulation pipette was filled with ACSF and placed close to either the proximal or distal dendrites of the CA1

pyramidal cell being recorded (see diagram in Fig. 3). In synaptic plasticity experiments, the recording and stimulation pipettes were filed with ACSF and the recording pipette was situated in the stratum radiatum (SR) of the CA1, whereas the stimulation pipette was placed in the SR of the CA3a. A ramp of stimulation intensities was applied to all slices to calculate a stimulation–output curve. A stimulation intensity resulting in a 50% maximal response amplitude was applied during the rest of the experiment. Stimuli were delivered at 0.06 Hz. For LTP induction, a baseline (20 min) was acquired to assess fEPSP stability and then either HFS (100 stimuli at 100 Hz applied 3 times, with 10 s intervals) or TBS (10 5 Hz bursts of 4 stimuli at 100 Hz applied twice, with 20 s interval) induction protocols were applied. After induction, the fEPSP was measured for 25 min, when the signal was stable. A depotentiation protocol (900 stimuli at 1 Hz) was then applied and fEPSPs were measured for 35 min All electrophysiological data were acquired using the Clampex 10.5 software (Axon Instruments) and analyzed using Clampfit 10.5 (Axon Instruments) and MiniAnalysis (Synaptosoft).

**Immunohistochemistry**. Tissue processing and immunochemistry were performed as previously[37,53]. Briefly, mice were sacrificed with an overdose of urethane injected intraperitoneally and they were perfused intracardially with ice-cold phosphate-buffered saline (PBS) followed by 4% paraformaldehyde (PFA) in PBS. The animal's brain was then further fixed in 4% PFA overnight and rinsed in PBS. Sagittal and parasagittal brain sections (40 μm) were obtained on a Leica VT1000 S vibratome. The sections were washed in PBS and blocked with 0.2% Triton X-100, 5% bovine serum albumin and 3% normal goat serum in PBS for 2 h. The slices were then incubated overnight at 4 °C with the primary antibody and then for 2 h at room temperature with the secondary antibody. After washing with PBS, the slices were transferred to glass slides, dried, and mounted with Vectashield with 4,6-diamidino-2-phenylindole (Vector Laboratories).

**Imaging**. Cell counting of interneurons in CA1, CA3, DG, BLA, and cortex was performed using the Neurolucida software (MBF Bioscience). Briefly, SOM+ and PV+ interneurons were identified on a fluorescence microscope by their immunohistochemical labeling. Sample images were obtained on a Zeiss LSM880 microscope.

**Behavioral tests**. Following animal handling (2–3 days), 2–4-month-old animals were subjected to behavioral testing by an experimenter blind to the animal's genotype. Both male and female animals were used in the experiments. Testing was performed during the light cycle, and the mice were tested in the following order: open field, NOR, and NOL tests. Two days after, the mice were tested in the elevated plus maze, and after 1 week, the mice underwent a fear memory test.

The open field test was performed as previously[53]. Briefly, mice were placed in a $50 \times 50$ cm$^2$ arena for 30 min with the illumination intensity set at 30 lux. The total distance traveled and the time (%) spent in the central ($30 \times 30$ cm$^2$) and peripheral area were measured offline, using the Smart Video Tracking System (Harvard Apparatus).

Assessment of anxiety levels was performed in an elevated plus maze as described[53]. The maze consisted of a cross-shaped metal structure with two open and illuminated arms and two opposing dark arms, enclosed by 30 cm high walls. Each arm was $50 \times 10$ cm and the four arms were separated by a square $10 \times 10$ cm central area. The mice were placed in the center of the arena and video tracked for 10 min The videos were analyzed offline to assess the total distance traveled in the maze and the time (%) spent in the different zones.

The NOR test was performed in the open field arena. On day 1, the mice were trained for 15 min with two identical objects placed near the periphery of the arena. One of the objects was then replaced by a different shape and color novel object 1 h after training, and the mice were allowed to explore the two objects for 15 min. Video tracking of the mice was analyzed offline by the experimenter to measure the time exploring each object. The mice were considered to be exploring if their head was directed at the object and they were sniffing it at a distance <1 cm. For training and the test session, a discrimination index (DI) was calculated using the equation:

$$DI = 100 \frac{NOE - FOE}{TE}$$

where NOE is the novel object exploration, FOE is the familiar object exploration, and TE is the total time of exploring the two objects[11].

The NOL test was also performed in the open field arena. On day 1, mice were trained for 15 min with two identical objects placed near the periphery of the arena. Visual cues were placed on the arena walls to allow the position of the objects to be identified, and 1 h after training, one of the objects was moved to a novel position and the mice were allowed to explore the two objects for 15 min. The object in the novel location of the 1 h test session was repositioned to a different novel location 24 h after training and the mice were again allowed to explore the objects for 15 min. The time of exploration was assessed as in the NOR test. For the training, 1-, and 24-h test, a DI was calculated using the equation:

$$DI = 100 \frac{NLE - FLE}{TE}$$

where NLE is the novel location exploration, FLE is the familiar location exploration, and TE is the total time exploring the two objects. We set a threshold

for object interaction in NOR and NOL tests. If animals explored objects <20 s during the training phase, the test was discontinued.

The fear conditioning test was performed in fear boxes (Panlab) controlled by the Freezing software (Panlab). One day before the training session, the mice were habituated for 3 h to the fear conditioning room. In the training session, the mice were introduced into the fear box and allowed to explore it for 4 min before being presented with a 80-db 28,000-Hz tone for 28 s (CS). Immediately afterwards, the mice were exposed to the same tone and a 0.5-mA foot shock for 2 s (US). The mice were kept in the fear box for 2 min and then returned to the home cage. Activity was measured with a piezoelectric grid at the bottom of the cage, and three activity levels were assessed: freezing, low, and high activities. The freezing time (%) before and after the shock was measured. To assess contextual fear memory, 1 day later the mice were placed for 4 min in the same cage but they were not exposed to the CS or the US, assessing freezing. After 5 h of rest, a cylindrical Plexiglas tube was introduced into the fear box to emulate a novel context. A novel odor (1% acetic acid) was sprayed into the box to avoid olfactory identification of the known context. The mice were introduced to the novel context and allowed to explore it for 2 min before presentation of the CS for 2 min to assess cued fear memory associated with the tone. Freezing was again measured before (to assess fear generalization) and during the presentation of the tone (to assess cued fear memory).

**Statistical methods**. Statistical analyses were performed using the SigmaStat package of SigmaPlot 12.5 software. All data were averaged and expressed as the mean ± s.e.m. in scatter plots. In box plots, the data are represented as the median and mean (black and white horizontal lines within the box, respectively), the first and third quartiles (bottom and top of the box, respectively), the 10th and 90th percentiles (whiskers above and below the box, respectively), and as raw data (colored dots over the boxplots). Student's $t$ test was used for statistical comparisons between two groups after checking for a normal distribution of the data. One-way analysis of variance (ANOVA) followed by Holm–Sidak post hoc test was used to compare the differences between the genotypes when data followed a normal distribution. If normality conditions were not fulfilled, we used ANOVA on ranks followed by Dunn's post hoc test. Both Holm–Sidak and Dunn's post hoc tests correct for multiple comparisons.

## Data availability

The data sets generated and/or analyzed during the current study are available from the corresponding author on request.

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

## Acknowledgements

We thank Dr. MI Aller and M. Linares for helping with mice genotyping. The authors gratefully acknowledge the financial support received from the Spanish Agency of Research (AEI) under the grant BFU2015-64656-R (to J.L.), co-financed by the European Regional Development Fund (ERDF), and the "Severo Ochoa" Program for Centres of Excellence in R&D (SEV-2013-0317 and SEV-2017-0723). This work was also supported by the Generalitat Valenciana through the program PrometeoII/2015/012 (to J.L.), the FPI fellowship program (to S.V. and A.G.), and the SyMBaD – Marie Curie ITN (agreement # 238608) (to W.M.).

## Author contributions

S.V., A.G., W.M. and A.V.P. performed research and analyzed data; J.L. designed research; and S.V. and J.L. wrote the paper.

## Competing interests

The authors declare no competing interests.
