## [Transparent Peer Review File · Nature Communications]

Reviewers' comments:

Reviewer #1 (Remarks to the Author):

The manuscript by Valbuena et al., reports on GABAergic transmission in the Ts2Cje mouse model of Down syndrome (DS). The authors found that although the number of GABAergic Parvalbumin-positive (PV) interneurons is unchanged, both spontaneous and miniature IPSCs are increased on CA1 principal neurons but unchanged in the amygdala. This phenotype is rescued by normalizing the dosage of the DS-triplicated gene *Grik1*, encoding for the GluK1 subunit of kainate receptors (KAR). The authors also show that *Girk1* overexpression affects GABAergic transmission mainly by changing release probability at GABAergic terminals. Furthermore, normalization of *Grik1* dosage rescues spatial memory deficits in the NOL test but did not rescue deficits in associative memory (Fear conditioning test), Novel object recognition (NOR test) and hyperactivity/reduced anxiety (open field/elevated plus tests). Surprisingly, the rescue of spatial memory was not paralleled by a concomitant rescue of CA1 synaptic plasticity. Therefore apparently ruling out a role for CA1 synaptic plasticity in this form of memory in DS animals.

To my knowledge, this is the first report showing a role for the triplicated gene *Grik1* in some aspects of DS pathology. Overall, the manuscript is well written, results are interesting, well documented, clearly presented and statistic is appropriate. The topic is potentially interesting for the Down syndrome field also considering that the role of GABAergic signaling in related cognitive deficits is still largely debated. However, a weakness of the study is the use of the Ts2Cje mouse model of DS. This mouse strain has been utilized in very few studies compared to the more widely used Ts65Dn or Dp(16)1Yey models. Since, the vast majority of studies assessing GABAergic deficits in DS have been done in these two later mouse strains it is difficult to compare the present results with the available literature. On the other hand, this is the first report assessing GABAergic dysfunctions in this mouse model and therefore a more in depth characterization is needed. Second point regards the narrative statements present in the abstract and throughout the manuscript that excessive synaptic inhibition is a feature of DS mouse model. This is possibly an over-simplistic and outdated view. Increased GABA transmission in DS mouse models seems to be region-specific (as also shown in the present manuscript) and paralleled by glutamatergic deficits. Indeed, increased GABA-mediated transmission has been only found in the dentate gyrus of Ts65Dn mice (Kleschevnikov 2004 and 2012), but not CA1 (Chakrabarti 2010; Best 2012; Parrini 2017). Moreover, the frequency of mIPSC was actually found decreased in the CA3 of Ts65Dn mice (Hanson 2007; Stagni 2013). Instead, decreased glutamatergic transmission (also described in the present manuscript) has been reported in hippocampal CA3 (Hanson 2007; Stagni 2013) and CA1 (Parrini 2017) of Ts65Dn mice. The few available data on human DS brain samples also do not clearly support over-inhibition. Therefore, the authors should cite and discuss appropriately the literature and perhaps substitute throughout the text "excessive synaptic inhibition" or similar statements with a more broad "altered excitatory/inhibitory balance" or analogous phrasing.

Major points:

1. The authors should clearly indicate in the title that the study has been performed in the Ts2Cje mouse model of DS. In addition, since *Grik1* triplication only affects spatial memory, perhaps a more appropriate title would be: "Unbalanced dendritic inhibition of CA1 neurons accounts for spatial memory deficits in the Ts2Cje mouse model of Down syndrome". Moreover, the authors should put more emphasis on the fact that restoring *Grik1* dosage only partially rescues cognitive phenotype in these mice and clearly state in the abstract that *Grik1* triplication does not play a role in associative memory, novel object recognition and CA1 synaptic plasticity deficits in these mice.
2. The authors show that the GluK1 agonist ATPA evoked larger currents in CA1 interneurons of Ts2Cje mice. However is not clear how where the neurons identified as GABAergic. Moreover, as the authors state that *Grik1* is normally only expressed in interneurons it would be interesting to evaluate possible misexpression due to triplication in Ts2Cje mice by IHC or in-situ hybridization. In addition, the authors could also evaluate the response of ATPA in CA1, CA3 principal neurons

and dentate granule cells of Ts2Cje mice to rule out any possible effects of Grik1 overexpression in glutamatergic neurons. Please also check if ATPA evoked currents are larger in the amygdala of Ts2Cje mice.

3. The GABAB-coupled inwardly rectifying potassium channels *Kcnj6* (*Kir3.2*) is also triplicated in DS mouse models. In fact, increased slow inhibitory currents through GABAB receptors have been reported in both CA1 and dentate neurons of Ts65Dn mice (Kleschevnikov 2012; Best 2012). In patch-clamp experiments, the authors have not used GABAB inhibitors during recordings. Were the slow GABAB currents excluded by the different kinetic during analysis?

4. The authors have evaluated the density of PV interneurons but not of other classes of interneurons. I think it is necessary to also evaluate at least the density of Somatostatin (SOM) interneurons in CA1, CA3, dentate gyrus and cortex to give a clearer picture of the GABAergic population in this strain of mice compared to the widely used Ts65Dn. Moreover, as the authors have also analyzed GABAergic transmission in the amygdala, the analysis of both PV and SOM neuronal density should also be extended to this area.

5. The authors report a decrease of mEPSC and an increase of mIPSC and sIPSC frequencies in the CA1 region of Ts2Cje mice. Are these defects also present in the other hippocampal subfields (CA3 and dentate gyrus) as seen in other DS mouse models? Are the alterations seen in miniature frequency paralleled by differences in synaptic densities as shown in Ts65Dn mice (Stagni 2013; Parrini 2017)? Evaluation of glutamatergic and GABAergic synaptic densities in the same areas will strongly improve the manuscript.

6. The results on bidirectional alterations of inhibitory synaptic transmission alterations in CA1 neurons is very interesting. However, is not clear from the method section where the stimulating electrode was placed for either apical or proximal stimulation (SR or OLM ?). Additionally, are the differences in release probability of GABAergic interneurons sufficient to explain the increase of mIPSC? Is release probability also affected in amygdala interneurons?

Moreover, as different classes of interneurons (SOM and PV) make synaptic contacts along the somatodendritic axis of CA1 pyramidal neurons, it is unclear how Grik1 triplication may affect in opposite ways release probability of these two classes of interneurons. In the discussion section, the authors suggest that this may be due to canonical and non-canonical signaling pathways. Experiments to explain such differences will increase the interest of the manuscript.

7. Regarding the synaptic plasticity experiments and the correlation with behavioral deficits, the results showing rescue of spatial memory performances without a concomitant improvement of LTP upon normalization of Grik1 expression are very interesting. The authors have fairly highlighted that these results indicate "that synaptic inhibition and spatial memory are independent from the LTP magnitude in the CA1 region of this model of DS". However, this is only true for spatial memory measured with the NOL test. Therefore, the authors should also highlight that LTP magnitude in the CA1 region may depend on decreased glutamatergic transmission and that it may still explain deficits in associative memory (contextual fear conditioning) and novel object recognition.

8. For the behavioral experiments is not clear whether the same mice were used in the different tests. Alternatively, in case the same animals performed more tests, what was the order of the tests?

9. Regarding decreased anxiety-related behaviors seen here in Ts2Cje animals, this phenotype appears somehow different from what previously reported for Ts65Dn mice, showing no difference (Martinez-Cue 2013, Vidal 2017) or increased anxiety-associated behavior (Lysenko 2014). This discrepancy must be clearly highlighted to the readers in the discussion.

10. Regarding the conclusions drawn from data presented in Supplementary Figure 9, these are not convincing to me. First, these are only correlation and (even if characterized by significant p value) do not imply a direct link between the two behaviors analyzed. Second, since these are correlations (NOT association as indicated by the authors) it cannot be stated that: "contextual fear conditioning is dominated by spatial memory only in the euploid condition" (also because the correlation in this case was not even significant; $p = 0.079$) or that "results suggest that reduced anxiety dominates fear responses in trisomic mice". These is clearly an over-interpretation of the data not supported by a solid experimental approach. Moreover, this interpretation is not even reinforced by changes of either mEPSCs or mIPSCs in the amygdala of Ts2Cje mice. Therefore,

unless the authors provide more convincing experimental data clearly linking alteration of contextual fear conditioning to spatial memory or reduced anxiety to fear responses I strongly encourage them to strongly dampen their conclusions.

Reviewer #2 (Remarks to the Author):

The manuscript of Valbuena et al characterizes the impact of the normalization of Grik1 dosage on the behavioral and physiological phenotypes present in the Ts2Cje mouse model of Down syndrome. They report that Grik1 normalization rescues memory impairments in this model and further reveal interesting bidirectional changes in the local inhibitory network of CA1. Overall the paper contains a well-presented and interesting set of data that will be useful to the field. I do have a several comments and questions that I think would help the authors improve the clarity of the manuscript and strengthen the conclusions they draw.

1. In Figure 1 the authors present behavioral data from the NOR and NOL assays. The change in novelty preference is clear in the Ts2Cje mice, however given the data in Figure 5 showing changes in OF exploration it is important to note if there is a difference between the groups in overall object exploration time both during the encoding phase and the test phase. It is not stated in the methods if a minimum duration of object exploration during encoding was necessary for a subject to be included. The obvious worry is that if a hyperactive mouse simply ignores the objects (for example, exploration <10sec across the 15min session) then the deficit in recall can be separable from a true memory deficit. The total interaction time for all tests and sessions should be compared across groups and included in the supplementary materials.

2. In Figure S1 they authors demonstrate the Grik1 mRNA levels in the Ts2Cje mice roughly reflect the increase in gene dosage, however the ATPA triggered currents in panel b shows that in CA1 interneurons the increase in KAR currents is much larger than 1.5x or 2x and seems closer to 8-10x. Can the authors explain this? This should be discussed.

3. In their characterization of the bidirectional modulation of GABA release by mIPSC amplitude the authors state in the text on p.6 "... a lower proportion of mIPSCs with amplitudes between 40 and 100 pA", yet in the figure the difference is characterized in the 40-80pA bin. This should be made consistent.

4. In general, I found the last section of the paper, dealing with the anxiety and fear phenotypes to be the weakest and most ambiguous part. The hyperactivity phenotype in Figure 5 was not rescued in the TsGrik1 +/- mice. The authors interpret this in terms of reduced anxiety, which is one possibility. However increased locomotion in a novel context could also reflect deficits in habituation and spatial coding, in fact, this is a classic phenotype of hippocampal lesions. It would be more convincing to look at the data in a temporal manner (exploration distance over time) to assess habituation in these animals compared to the Ts2Cje mice. The same can be said for the fear deficits. Given the significant decrease in fear memory in both the context and tone test I agree it is valid to postulate defects in the amygdala circuit, however it does not rule out contributions from the hippocampus that are not rescued in the TsGrik1 +/- mice. This possibility must be noted.

5. The freezing levels reported in the controls are extremely high for the protocol used (1 shock, 0.5mA, 2") and not consistent with the vast majority of the literature. Please expand the methods sections on how freezing was scored.

6. The correlation analyses in Figure S9 is not convincing or useful. The findings in the euploid mice (panel a) are not significant and due to the surprisingly high levels of freezing creating a ceiling effect, the correlation doesn't say much about the relationship between these measures. I

think the paper would be stronger with this entire section and figure simply eliminated.

Reviewer #3 (Remarks to the Author):

The study by Valbuena et al. investigates the behavioral and electrophysiological consequences of GluK1 disruption in the Ts2Cje mouse model of Down's syndrome. In addition to comparing euploid controls and triploid mutants, the authors also look at trisomic Ts2Cje mice that are euploid for the Grik1 gene. While the general question of how GluK1 regulates brain function is interesting, both in normal and disease conditions, this work does very little to enhance our understanding of the links between genes, cellular activity, and behavior.

1. While behavioral rescue (of some but not all assays) through normalization of Grik1 dosage is notable, there is no circuit specificity to the analyses. Are these mice altered in sensory, emotional, or motor function? To what extent do the observed phenotypes arise from the hippocampal cellular changes reported later in the manuscript?

2. The major electrophysiological conclusion is that altered Grik1 expression differentially impacts inhibition along the somatodendritic arbor of CA1 pyramidal cells. This conclusion is based wholly on altered mIPSC amplitudes and kinetics, which are purported to reflect dendritic localization. However, a wide range of mechanisms influence both these properties (e.g., receptor subunit composition, receptor density, presynaptic release properties – synchronous versus asynchronous release, and postsynaptic conductances that modify the membrane time constant). None of these additional factors is explored here, making it impossible to support the primary conclusions.

3. The plasticity data seem completely out of place – the lack of rescue upon normalizing Grik1 levels is interesting but seems disconnected from the rest of the study, which is about changes that are Grik1-dependent.

4. Some minor issues: PV immunostaining for cell density is very hard to interpret given the activity-dependence of PV expression. All statistical tests should be named in the text, with n- and exact p-values provided. No mention is made of corrections for multiple comparisons throughout, for example, in Figure 2a and c. Indeed, these analyses would better be done with a distribution test (e.g., k-s) where arbitrary data binning is not required.

Point-by-point answer to the reviewers

We thank very much the thoughtful evaluation of our paper by the three referees. We hope to have addressed all the referees' demands and thank them for all the insights.

Reviewer #1 (Remarks to the Author):

The manuscript by Valbuena et al., reports on GABAergic transmission in the Ts2Cje mouse model of Down syndrome (DS). The authors found that although the number of GABAergic Parvalbumin-positive (PV) interneurons is unchanged, both spontaneous and miniature IPSCs are increased on CA1 principal neurons but unchanged in the amygdala. This phenotype is rescued by normalizing the dosage of the DS-triplicated gene *Grik1*, encoding for the GluK1 subunit of kainate receptors (KAR). The authors also show that *Girk1* overexpression affects GABAergic transmission mainly by changing release probability at GABAergic terminals. Furthermore, normalization of *Grik1* dosage rescues spatial memory deficits in the NOL test but did not rescue deficits in associative memory (Fear conditioning test), Novel object recognition (NOR test) and hyperactivity/reduced anxiety (open field/elevated plus tests). Surprisingly, the rescue of spatial memory was not paralleled by a concomitant rescue of CA1 synaptic plasticity. Therefore apparently ruling out a role for CA1 synaptic plasticity in this form of memory in DS animals.

To my knowledge, this is the first report showing a role for the triplicated gene *Grik1* in some aspects of DS pathology. Overall, the manuscript is well written, results are interesting, well documented, clearly presented and statistic is appropriate. The topic is potentially interesting for the Down syndrome field also considering that the role of GABAergic signaling in related cognitive deficits is still largely debated. However, a weakness of the study is the use of the Ts2Cje mouse model of DS. This mouse strain has been utilized in very few studies compared to the more widely used Ts65Dn or Dp(16)1Yey models. Since, the vast majority of studies assessing GABAergic deficits in DS have been done in this two later mouse strains is difficult to compare the present results with the available literature. On the other hand, this is the first report assessing GABAergic dysfunctions in this mouse model and therefore a more in depth characterization is needed.

We thank the reviewer for thinking that our results are interesting and well documented. We agree with the reviewer regarding the fact that the Ts2Cje model has been less characterized than Ts65Dn counterpart. However, we must indicate that the Ts2Cje model appeared from a chromosomal rearrangement of the triplicated segment present in Ts65Dn mice and therefore both models contain the same set of triplicated genes. The only difference between these models is the translocation of the extra MMU16 segment into MMU12 in Ts2Cje mice. This translocation favors the transmission of the trisomy to the progeny, and constitutes the reason why we chose Ts2Cje over Ts65Dn mice. As the reviewer states, it is totally true that Ts65Dn (and, to a lesser extent, Dp(16)1Yey), constitutes the most widely used model of DS. However, to the best of our knowledge, all available studies using Ts2Cje mice have reported the presence of the same phenotypic alterations reported in Ts65Dn mice. Thus, it has been shown that both Ts65Dn and Ts2Cje mice present reduced density of dendritic spines in DG (Kleschevnikov et al., 2004; Villar et al., 2005), increased oxidative stress (Ishihara et al., 2009; Corrales et al., 2017), impaired neurogenesis (Ishihara et al., 2010; Clark et al. 2006; Chakrabarti et al. 2007), deficits in spatial memory and LTP (Reeves et al., 1995; Siarey et al., 1997; Kaur et al., 2014) and Alzheimer-like phenotypes (Holtzman et al., 1996; Granholm et al., 2000; Jiang et al., 2016).

Conversely, our data further evidence such similarity, as we found altered GABAergic and glutamatergic transmission in the CA1 region of Ts2Cje mice, impaired performance in the NOR tasks and deficient anxiety and fear responses (as previously reported in Ts65Dn animals by Costa et al., 2008; Chakrabarti et al., 2010; Sichiri et al., 2011; García-Cerro et al., 2014; Parrini et al., 2017). Interestingly, the only remarkable difference that we found in Ts2Cje animals in comparison to previous studies in Ts65Dn mice is the absence of increased density of interneurons (SOM⁺ and PV⁺ cells) in CA1, CA3, DG, BLA or neocortex, a result that better matches data found from postmortem tissue of DS patients (Ross et al., 1984; Kobayashi et al., 1990). In conclusion, the great degree of similarity between both models suggest that Ts65Dn and Ts2Cje mice may be considered equivalent mouse models of DS, and there is no indication pointing any important difference between them. Finally, regarding the GABAergic dysfunctions in our mouse model, we believe that our work constitutes a deep scrutiny of alterations in inhibitory systems in the CA1 region of the hippocampus. Indeed, we describe a Grik1-dependent unbalance in GABAergic activity over CA1 pyramidal cell's somatodendritic axis, which provides the first mechanistic explanations for the spatial memory impairment described in DS mouse models. Finally, we show that the genetic normalization of Grik1 dosage was enough to normalize these deficits. However, we understand the reviewer's concern regarding the characterization of GABAergic alterations in these animals and, as stated below (point 5) and, in response we have evaluated excitatory and inhibitory activity in the CA3 and DG regions (supplementary fig 8), essentially confirming previous findings.

Second point regards the narrative statements present in the abstract and throughout the manuscript that excessive synaptic inhibition is a feature of DS mouse model. This is possibly an over-simplistic and outdated view. Increased GABA_A transmission in DS mouse models seems to be region-specific (as also shown in the present manuscript) and paralleled by glutamatergic deficits. Indeed, increased GABA_A-mediated transmission has been only found in the dentate gyrus of Ts65Dn mice (Kleschevnikov 2004 and 2012), but not CA1 (Chakrabarti 2010; Best 2012; Parrini 2017). Moreover, the frequency of mIPSC was actually found decreased in the CA3 of Ts65Dn mice (Hanson 2007; Stagni 2013). Instead, decreased glutamatergic transmission (also described in the present manuscript) has been reported in hippocampal CA3 (Hanson 2007; Stagni 2013) and CA1 (Parrini 2017) of Ts65Dn mice. The few available data on human DS brain samples also do not clearly support over-inhibition. Therefore, the authors should cite and discuss appropriately the literature and perhaps substitute throughout the text "excessive synaptic inhibition" or similar statements with a more broad "altered excitatory/inhibitory balance" or analogous phrasing.

We thank the reviewer for rising this point. It is true that previous reports described a reduced inhibitory activity in the CA3 region of the hippocampus of Ts65Dn mice (Hanson 2007; Stagni 2013). However, we must highlight the fact that almost every other study evaluating GABAergic function (and related behavioral alterations) reported an overall increased inhibition in the hippocampus (e.g. Kleschevnikov et al., 2004; 2012a; 2012b; Fernández et al., 2007; Chakrabarti et al., 2010; Martínez-Cué et al., 2013; etc.). These reports have crystalized in the so called GABAergic hypothesis (reviewed in Fernandez and Gardner, 2007; Contestabile et al., 2017; Zorrilla de San Martín et al., 2018). Nevertheless, we understand the reviewer's statement regarding the use of "excessive synaptic inhibition" and have minimized using this term in the text

Major

points:

1. The authors should clearly indicate in the title that the study has been performed in the Ts2Cje mouse model of DS. In addition, since Grik1 triplication only affects spatial memory, perhaps a more appropriate title would be: "Unbalanced dendritic inhibition of CA1 neurons accounts for spatial memory deficits in the Ts2Cje mouse model of Down syndrome". Moreover, the authors should put more emphasis on the fact that restoring Grik1 dosage only partially rescues cognitive phenotype in these mice and clearly state in the abstract that Grik1 triplication does not play a role in associative memory, novel object recognition and CA1 synaptic plasticity deficits in these mice.

We thank the reviewer for the suggestion. We find it adequate to clearly state in the title that the study links unbalanced inhibition and spatial memory deficits in the Ts2Cje model and we have changed the title accordingly. Besides, given that our main idea is to precisely define the roles of Grik1 triplication in deficits present in these animals, we find totally adequate to state in the abstract that Grik1 triplication plays a role in spatial memory deficits but not in the ones described by the reviewer, and we wanted to change the abstract accordingly. We examined the abstract and given the limited number of words allowed, found difficult to mention so many aspects as the referee suggests. So, we think that leaving the sentence "not the changes in synaptic plasticity or the other behavioral modifications observed" would be enough for the abstract.

2. The authors show that the GluK1 agonist ATPA evoked larger currents in CA1 interneurons of Ts2Cje mice. However is not clear how where the neurons identified as GABAergic. Moreover, as the authors state that Grik1 is normally only expressed in interneurons it would be interesting to evaluate possible misexpression due to triplication in Ts2Cje mice by IHC or in-situ hybridization. In addition, the authors could also evaluate the response of ATPA in CA1, CA3 principal neurons and dentate granule cells of Ts2Cje mice to rule out any possible effects of Grik1 overexpression in glutamatergic neurons. Please also check if ATPA evoked currents are larger in the amygdala of Ts2Cje mice.

We thank the reviewer for bringing that point to our attention. We didn't include any statement on how we identified GABAergic cells for ATPA perfusion experiments due to the fact that CA1 interneurons can easily be identified by the location of their somas. Therefore, interneurons were unequivocally identified as cells which had their soma clearly out of the pyramidal cell layer. Nevertheless, we have indicated this in the Methods section in order to clarify the strategy followed for interneuron identification (see page 16)

Regarding the expression of Grik1, we agree with the reviewer on the point that it would be interesting to determine whether there is some degree of misexpression in the hippocampus. To evaluate this point, and taking into account that Grik1 is mostly expressed in interneurons in CA1, we performed recordings of CA1, CA3 and DG principal cells applying ATPA to address the presence of GluK1 in these cells. Results compellingly showed the absence of Grik1 misexpression (new Supplementary fig 1c-e). Regarding the amygdala region, although we agree with the reviewer in that it may be interesting to perform a detailed evaluation of the cell types expressing Grik1 in control and trisomic conditions, we believe that this characterization is beyond the scope of this work. It is already known that KARs are expressed in BLA principal cells and interneurons (Braga et al., 2009; Aroniadou-Anderjaska et al., 2012). A further characterization would be of interest but, given the lack of Grik1 dependent effects in behavioral

phenotypes related to the amygdala function (i.e. anxiety and fear) and BLA physiology (mEPSCs and mIPSCs frequency, amplitude and charge transfer), we believe that it would add little to the main point of the paper.

3. The GABAB-coupled inwardly rectifying potassium channels *Kcnj6* (*Kir3.2*) is also triplicated in DS mouse models. In fact, increased slow inhibitory currents through GABAB receptors have been reported in both CA1 and dentate neurons of Ts65Dn mice (Kleschevnikov 2012; Best 2012). In patch-clamp experiments, the authors have not used GABAB inhibitors during recordings. Were the slow GABAB currents excluded by the different kinetic during analysis?

We agree with the reviewer in highlighting the role of *Kcnj6* triplication in DS mouse models – especially the Ts65Dn-. It is totally true that we did not apply GABAB antagonists during our recordings. However, it is not necessary to exclude GABAB currents during the analysis of m/sIPSCs simply because the extremely slow –in the range of hundreds of milliseconds- K⁺ currents evoked by GABAB activation could not be confused with m/sIPSCs –which present a rise and decay times more than an order of magnitude faster-. Indeed, in the paper by Best et al (2012), the authors describe the presence of electrical stimulation-evoked GABAB currents, in CA1 pyramidal cells, with an onset of activation slower than 500 ms, whereas we are presenting data regarding mIPSCs with a rise time faster than 10 ms.

4. The authors have evaluated the density of PV interneurons but not of other classes of interneurons. I think it is necessary to also evaluate at least the density of Somatostatin (SOM) interneurons in CA1, CA3, dentate gyrus and cortex to give a clearer picture of the GABAergic population in this strain of mice compared to the widely used Ts65Dn. Moreover, as the authors have also analyzed GABAergic transmission in the amygdala, the analysis of both PV and SOM neuronal density should also be extended to this area.

We totally agree with the reviewer regarding this point. We evaluated the density of PV⁺ interneurons in the CA1, CA3 and neocortex, showing that this parameter was unaltered across genotypes. We are aware that different reports (e.g. Chakrabarti et al., 2010) have previously reported that Ts65Dn mice present increased density of SOM⁺ cells. Therefore, in order to further characterize our animals, we have performed the analysis proposed by the reviewer, evaluating the density of PV⁺ and SOM⁺ cells in CA1, CA3, DG, cortex and amygdala (Supplementary figs 4 and 5).

5. The authors report a decrease of mEPSC and an increase of mIPSC and sIPSC frequencies in the CA1 region of Ts2Cje mice. Are these defects also present in the other hippocampal subfields (CA3 and dentate gyrus) as seen in other DS mouse models? Are the alterations seen in miniature frequency paralleled by differences in synaptic densities as shown in Ts65Dn mice (Stagni 2013; Parrini 2017)? Evaluation of glutamatergic and GABAergic synaptic densities in the same areas will strongly improve the manuscript.

We thank the reviewer for rising this point. We were convinced that it would be very interesting to expand our analysis of basal excitatory and inhibitory function (EPSCs and IPSCs) in the CA1 region, and we have performed these analyses in other regions in the hippocampal formation (CA3 and DG) (see Supplementary fig 8). Regarding the evaluation of the glutamatergic and GABAergic synaptic density, although such type of anatomical analyses would be interesting to

perform, we believe that the carried out functional analysis (i.e. mIPSCs and mEPSCs) already provides information on this issue.

6. The results on bidirectional alterations of inhibitory synaptic transmission alterations in CA1 neurons is very interesting. However, is not clear from the method section where the stimulating electrode was placed for either apical or proximal stimulation (SR or OLM ?). Additionally, are the differences in release probability of GABAergic interneurons sufficient to explain the increase of mIPSC? Is release probability also affected in amygdala interneurons? Moreover, as different classes of interneurons (SOM and PV) make synaptic contacts along the somatodendritic axis of CA1 pyramidal neurons, it is unclear how Grik1 triplication may affect in opposite ways release probability of these two classes of interneurons. In the discussion section, the authors suggest that this may be due to canonical and non-canonical signaling pathways. Experiments to explain such differences will increase the interest of the manuscript.

We thank the reviewer for presenting this concern. We show in the cartoon on Fig. 3 the position of the electrodes for proximal and distal stimulation. +We have indicated this fact in the Methods section (page 16). Regarding the mechanistic basis for the bidirectional action of GluK1-containing KARs, further experiments may be performed in order to determine the involvement of each of the signaling pathways in the promotion or decrease of GABA release by interneurons. However, such an evaluation of the signaling mechanisms underlying KAR-mediated control of GABA release has previously been performed. Our group previously showed that the KAR-mediated reduction in GABA release from CA1 interneurons was caused by the activation of a metabotropic pathway (Rodríguez-Moreno and Lerma, 1998), a result that has been further corroborated by other laboratories (e.g. Cunha et al., 2000, Daw et al., 2010). Besides, different papers showed that KAR-mediated promotion of GABA release in CA1 occurred through the activation of an ionotropic pathway (Cossart et al., 2001; Jiang et al., 2001). Finally, there is a beautiful paper showing that KAR promotion of GABA release occurred through the ionotropic pathway, whereas KAR-mediated inhibition of GABA release depended on the metabotropic activity (Bonfardin et al., 2010; see also fig 3 in Lerma and Marques, 2013). Therefore, in light of the literature, we may speculate that Grik1-dependent unbalance in inhibition would occur taking advantage of both signaling pathways (with the ionotropic pathway promoting GABA release over distal dendrites and metabotropic one reducing GABA release over proximal dendrites). The aim of the paper was to show the effects of Grik1 triplication in the physiological and behavioral deficits present in Ts2Cje mice, rather than to find out what KAR signaling mechanism was involved. In order to avoid overstatements regarding these signaling mechanistic basis –which would be better addressed in future work- we maintain the question open with the following statement in the discussion “*In the hippocampus, GluK1 is present in GABAergic terminals, where it may bidirectionally regulate GABA release through its canonical and non-canonical signaling pathways*”.

7. Regarding the synaptic plasticity experiments and the correlation with behavioral deficits, the results showing rescue of spatial memory performances without a concomitant improvement of LTP upon normalization of Grik1 expression are very interesting. The authors have fairly highlighted that these results indicate “that synaptic inhibition and spatial memory are independent from the LTP magnitude in the CA1 region of this model of DS”. However, this is only true for spatial memory measured with the NOL test. Therefore, the authors should also highlight that LTP magnitude in the CA1 region may depend on decreased glutamatergic

transmission and that it may still explain deficits in associative memory (contextual fear conditioning) and novel object recognition.

We thank the reviewer for acknowledging the interest regarding the lack of correlation between spatial memory impairment and LTP deficits in the CA1 region. However, the LTP deficits may still correlate with Grik1-independent behavioral deficits, such as NOR and associative memory. We thank the reviewer for raising this point and we have changed the text accordingly to clearly state this issue. On the other hand, regarding the causes underlying reduced LTP in Ts2Cje and TsGrik1+/- mice, we agree with the reviewer in the possible involvement of alterations in glutamatergic transmission (namely mEPSC frequency), and we have incorporated this possibility in the Discussion (see pag 12 , end of 1st para)

8. For the behavioral experiments is not clear whether the same mice were used in the different tests. Alternatively, in case the same animals performed more tests, what was the order of the tests?

We agree with the reviewer in that it is important to state that the animals were used in different test, and to inform on the order of experiments followed. We, have made it clearer at several places (e.g. page 10, upper lines), including in the Methods section (page 17).

9. Regarding decreased anxiety-related behaviors seen here in Ts2Cje animals, this phenotype appears somehow different from what previously reported for Ts65Dn mice, showing no difference (Martinez-Cue 2013, Vidal 2017) or increased anxiety-associated behavior (Lysenko 2014). This discrepancy must be clearly highlighted to the readers in the discussion.

We agree with the reviewer on this point. We believe that it will be important to highlight the discrepancies between Ts2Cje and Ts65Dn animals in order to better establish the characterization of both models. However, the literature on this issue is really controversial. Thus, as the reviewer states, it has been reported that Ts65Dn mice may present either increased or unaltered anxiety-like responses in the open field arena. However, Ts65Dn animals have been reported to stay more time in the open arms of the elevated plus maze (Shichiri et al., 2011; Corrales et al., 2013), suggesting a reduction in anxiety-like phenotypes. Our data show that Ts2Cje mice present reduced anxiety-like phenotypes in both open field and elevated plus maze tests. Therefore, it seems that the discrepancies previously reported would affect Ts65Dn –which would present increased anxiety in the open field and reduced anxiety in the elevated plus maze. In any case, we agree with the reviewer regarding the importance of highlighting the discrepancies between Ts65Dn and Ts2Cje. However, given the space constriction and the specificity of the , we leave to the expert readers to note this aspect and get their own conclusions.

10. Regarding the conclusions drawn from data presented in Supplementary Figure 9, these are not convincing to me. First, these are only correlation and (even if characterized by significant p value) do not imply a direct link between the two behaviors analyzed. Second, since these are correlations (NOT association as indicated by the authors) it cannot be stated that: “contextual fear conditioning is dominated by spatial memory only in the euploid condition” (also because the correlation in this case was not even significant; $p = 0.079$) or that “results suggest that reduced anxiety dominates fear responses in trisomic mice”. These is clearly an over-interpretation of the data not supported by a solid experimental approach. Moreover, this

interpretation is not even reinforced by changes of either mEPSCs or mIPSCs in the amygdala of Ts2Cje mice. Therefore, unless the authors provide more convincing experimental data clearly linking alteration of contextual fear conditioning to spatial memory or reduced anxiety to fear responses I strongly encourage them to strongly dampen their conclusions.

We also agree on the fact that the data provided by supplementary Fig. 9 are merely informative, rather than demonstrative. We thank the reviewer for rising this issue, and believe it would be important to restrict the conclusions obtained from such analyses. Regarding this point, we have modified the text in order to explain that the performance in the NOL test was not correlated at all in trisomic (Ts2Cje + TsGrik1+/-) animals with time freezing in the fear conditioning tests. In contrast, we found a significant inverse correlation between anxiety levels and fear responses in these animals, suggesting that anxiety may be playing a role in such fear responses. We believe this is an important indication for not getting strong conclusions out of these tests, which may be valuable for this and future studies (see page 10, end of 1st para).

Reviewer #2 (Remarks to the Author):

The manuscript of Valbuena et al characterizes the impact of the normalization of Grik1 dosage on the behavioral and physiological phenotypes present in the Ts2Cje mouse model of Down syndrome. They report that Grik1 normalization rescues memory impairments in this model and further reveal interesting bidirectional changes in the local inhibitory network of CA1. Overall the paper contains a well-presented and interesting set of data that will be useful to the field. I do have a several comments and questions that I think would help the authors improve the clarity of the manuscript and strengthen the conclusions they draw.

1. In Figure 1 the authors present behavioral data from the NOR and NOL assays. The change in novelty preference is clear in the Ts2Cje mice, however given the data in Figure 5 showing changes in OF exploration it is important to note if there is a difference between the groups in overall object exploration time both during the encoding phase and the test phase. It is not stated in the methods if a minimum duration of object exploration during encoding was necessary for a subject to be included. The obvious worry is that if a hyperactive mouse simply ignores the objects (for example, exploration <10sec across the 15min session) then the deficit in recall can be separable from a true memory deficit. The total interaction time for all tests and sessions should be compared across groups and included in the supplementary materials.

We understand the reviewer's worries regarding the analysis and interpretation of NOR and NOL assays. It is true that a reduction in the exploration time may produce deficits in exploration and, subsequently, impaired discrimination of the novel object identity or location. Accordingly, in order ensure meaningful object exploration tests, we set a threshold for object interaction (20s) during analysis. As we stated in the Reporting Summary, if a mouse didn't pass that threshold (something that happened very few times) we discontinued the analysis. We thank the reviewer for raising the point, indeed, and we have included a specific statement on the establishment of such a threshold (page 18). In addition, the Supplementary fig 6a and 6b, show the discrimination index for the training phase (adquisition) of NOL and NOR, respectively, demonstrating that mice from all genotypes explored equally the two identical objects placed in training locations.

2. In Figure S1 they authors demonstrate the Grik1 mRNA levels in the Ts2Cje mice roughly

reflect the increase in gene dosage, however the ATPA triggered currents in panel b shows that in CA1 interneurons the increase in KAR currents is much larger than 1.5x or 2x and seems closer to 8-10x. Can the authors explain this? This should be discussed.

We thank the reviewer for rising this concern. It is true that, whether the triplication of Grik1 resulted in a 1.5/2-fold increase in mRNA expression in Ts2Cje mice, the ATPA-evoked currents in interneurons showed a much larger increase. This is the expected consequence of introducing GluK1 subunits in KARs which, under control conditions, lack this subunit. As our group and others have previously reported, the KARs in interneurons are diverse, with KARs presenting and lacking the GluK1 subunit (Christensen et al., 2004; Wyeth et al., 2017, etc.). We applied a concentration of ATPA (1 μ M) which only activates GluK1-containing receptors but no other combinations of KAR subunits. The activated currents are above the observed 1.5/2-fold increase in Grik1 mRNA in Ts2Cje mice likely due to the fact that there is no linear correlation between the amount of mRNA coding for GluK1 and the amount of GluK1 protein.

3. In their characterization of the bidirectional modulation of GABA release by mIPSC amplitude the authors state in the text on p.6 "... a lower proportion of mIPSCs with amplitudes between 40 and 100 pA", yet in the figure the difference is characterized in the 40-80pA bin. This should be made consistent.

We thank the reviewer for highlighting this typo. The correct statement is "a lower proportion of mIPSCs with amplitudes between 40 and 80 pA" and we have changed the text accordingly.

4. In general, I found the last section of the paper, dealing with the anxiety and fear phenotypes to be the weakest and most ambiguous part. The hyperactivity phenotype in Figure 5 was not rescued in the TsGrik1+/- mice. The authors interpret this in terms of reduced anxiety, which is one possibility. However increased locomotion in a novel context could also reflect deficits in habituation and spatial coding, in fact, this is a classic phenotype of hippocampal lesions. It would be more convincing to look at the data in a temporal manner (exploration distance over time) to assess habituation in these animals compared to the Ts2Cje mice. The same can be said for the fear deficits. Given the significant decrease in fear memory in both the context and tone test I agree it is valid to postulate defects in the amygdala circuit, however it does not rule out contributions from the hippocampus that are not rescued in the TsGrik1+/- mice. This possibility must be noted.

We agree with the reviewer regarding this point. It is true that the hyperactivity phenotype may be explained in many ways, one of them being the reduced anxiety, another being the hippocampal alterations. We would like to point out that our intention was not to link hyperactivity phenotypes to anxiety in trisomic animals. It has been previously shown that other mouse models (e.g. Ts65Dn) present increased locomotor activity in different tests (Braudeau et al., 2011; Kleschevnikov et al., 2012a). By reporting the hyperlocomotion phenotype present in Ts2Cje animals we just wanted to highlight the existence of another similarity with Ts65Dn mice, without evaluating its underlying possible explanations (given it was not restored upon Grik1 normalization).

5. The freezing levels reported in the controls are extremely high for the protocol used (1 shock, 0.5mA, 2") and not consistent with the vast majority of the literature. Please expand the methods sections on how freezing was scored.

We are aware of the high level of freezing shown by euploid animals. We believe that this may be strain-specific. Our results show that our animals presented ~70% time freezing after the shock and this freezing level was actually caused by the shock (0.5 mA, 2''), since we controlled the time the animals froze once placed in the box and it was always less than 15%.

6. The correlation analyses in Figure S9 is not convincing or useful. The findings in the euploid mice (panel a) are not significant and due to the surprisingly high levels of freezing creating a ceiling effect, the correlation doesn't say much about the relationship between these measures. I think the paper would be stronger with this entire section and figure simply eliminated.

We understand the reviewer's concern regarding the information provided by the Fig. S9. It is true that the high level of freezing may create a ceiling effect, making it difficult to evaluate or interpret the correlations that we provide. Our intention when including such analyses in the manuscript was to highlight the correlation between anxiety levels and fear responses in trisomic (but not euploid) mice. In order to avoid overstatements, as suggested by the referee 1, we have changed the text to make it clear that our correlations are merely informative and only try to provide a note of caution at the time of interpreting fear memory tests under different anxiety scenarios. Alternatively, we are open to remove this supplementary figure and the accompanying text, but as mentioned above, we find it useful for future studies.

Reviewer #3 (Remarks to the Author):

The study by Valbuena et al. investigates the behavioral and electrophysiological consequences of GluK1 disruption in the Ts2Cje mouse model of Down's syndrome. In addition to comparing euploid controls and triploid mutants, the authors also look at trisomic Ts2Cje mice that are euploid for the Grik1 gene. While the general question of how GluK1 regulates brain function is interesting, both in normal and disease conditions, this work does very little to enhance our understanding of the links between genes, cellular activity, and behavior.

We thank the reviewer for the thoughtful revision of our manuscript. We agree on the idea that our work improves our knowledge on the role of Grik1 in normal and disease conditions. However, we think that our research –especially due to our genetic normalization approach– really does contribute to expanding our understanding on the important link between gene triplication and physiological/behavioral deficits present in DS models. Indeed, our work evaluates the specific involvement of the Grik1 in such deficits, dissecting the participation of this gene in a plethora of cellular physiological and behavioral alterations present in the Ts2Cje model of DS. Whereas most of the studies regarding DS have been performed using pharmacological tools, we chose a precise gene normalization approach, similar to the ones revealing the involvement of Kcnj6 in NOR and DG LTP deficits (Kleschevnikov et al., 2017) or the participation of Dyrk1a in Alzheimer disease-like phenotypes present in Ts65Dn mice (García-Cerro et al., 2017). Even though we don't ignore the complexity behind DS phenotypes, we firmly believe that the use of such genetic approaches constitutes a vast improvement over pharmacological tools, providing precise information on the genes which, when triplicated, underlie well known alterations at higher levels.

1. While behavioral rescue (of some but not all assays) through normalization of Grik1 dosage is

notable, there is no circuit specificity to the analyses. Are these mice altered in sensory, emotional, or motor function? To what extent do the observed phenotypes arise from the hippocampal cellular changes reported later in the manuscript?

We thank the reviewer for raising this point, and agree on the fact that analyses on sensory or motor function would improve the manuscript. Accordingly, we have performed such behavioral analyses and include them in the study (Supplementary fig. 3). Besides, regarding the behavioral-physiological analyses, we have to point out that our genetic normalization approach not only resulted in a rescue of the spatial memory deficits present in the Ts2Cje model, but also in a recovery of alterations in the inhibitory deficits present in the hippocampal CA1 region. Indeed, we describe a novel unbalance in dendritic inhibition of CA1 pyramidal neurons that, for the best of our knowledge, provides with the first mechanistic explanation accounting for the observed deficits (and recovery) of spatial memory in a mouse model of DS. We think that our conclusions are well supported not only by our results, but also by the increasing evidences pointing to a crucial role of distal dendritic inhibition in hippocampal function and associated memory processes (Leão et al., 2012, Siwani et al., 2017, Turi et al., 2019, etc.). However, we are conscious about the fact that an increased exploration of hippocampal physiology alterations would further support our conclusions, and have performed mEPSCs and mIPSCs analyses in other regions (CA3 and DG) of the hippocampus in order to determine whether different systems may also participate in the observed behavioral deficits (see Supplementary fig 8)

2. The major electrophysiological conclusion is that altered Grik1 expression differentially impacts inhibition along the somatodendritic arbor of CA1 pyramidal cells. This conclusion is based wholly on altered mIPSC amplitudes and kinetics, which are purported to reflect dendritic localization. However, a wide range of mechanisms influence both these properties (e.g., receptor subunit composition, receptor density, presynaptic release properties – synchronous versus asynchronous release, and postsynaptic conductances that modify the membrane time constant). None of these additional factors is explored here, making it impossible to support the primary conclusions.

We thank the reviewer for raising this point. Part of our manuscript conclusions lay on an in-depth analysis of m/sIPSCs events. We agree in that the properties of mIPSC events may be affected by different mechanisms. We must highlight, however, that such analysis was further corroborated by studying the PPR of evoked responses (which evaluate presynaptic release properties). Thus, whether mIPSC analyses provided evidences about asynchronous release, PPR experiments (performed by electrically stimulating GABA release from distal and proximal inhibitory synapses; see Fig 3) informed about synchronous GABA release. We must also note that, although KARs have repeatedly been reported to modulate GABA release in a bidirectional manner (Rodríguez-Moreno and Lerma, 1998; Cunha et al., 2001; Mulle et al., 2000; Cossart et al., 2001; Jiang et al., 2001; Bonfardin et al., 2010), no report has ever described any KAR mediated effect on subunit composition or postsynaptic conductances of GABA receptors. Indeed, such effects would be extremely unlikely due to the fact that GluK1-containing KARs present negligible levels of expression in CA1 pyramidal cells (Paternain et al., 2000; see Supplementary fig 1c-e too). Therefore, we think that our interpretations rest on the concurrent conclusions obtained by the different experimental approaches performed in this work and the wide literature on the field of KAR-mediated regulation of GABA responses.

3. The plasticity data seem completely out of place – the lack of rescue upon normalizing Grik1

levels is interesting but seems disconnected from the rest of the study, which is about changes that are Grik1-dependent.

We thank the reviewer for rising this point. We believe, however, that this is one of the most important outcomes of the manuscript: Grik1 normalization did not restore LTP deficits in the CA1 region of the hippocampus. Indeed, most of the literature regarding synaptic plasticity deficits in DS has reported that these deficits were restored upon pharmacological blockade of inhibitory activity, and the general conclusion in the field has been that inhibitory synaptic enhancement underlies synaptic plasticity deficits in this region (Costa and Grybko et al., 2005; Fernandez et al., 2007; Martínez-Cué et al., 2013). Our work implies that this assumption may not be correct. Thus, our genetic normalization strategy was sufficient to precisely restore the inhibitory synaptic deficits observed in the CA1 region of Ts2Cje mice but not the synaptic plasticity deficits observed in this region. These results suggest, therefore, that it is plausible to state that inhibitory synaptic alterations –caused by Grik1 triplication, as we demonstrated- in the CA1 region are not paralleled by synaptic plasticity deficits –independent on Grik1-. We believe that this result will stimulate the field to further switch to genetic-based approaches in the pursue of a mechanistic explanation of synaptic plasticity deficits in DS mouse models.

4. Some minor issues: PV immunostaining for cell density is very hard to interpret given the activity-dependence of PV expression. All statistical tests should be named in the text, with n- and exact p-values provided. No mention is made of corrections for multiple comparisons throughout, for example, in Figure. 2a and c. Indeed, these analyses would better done with a distribution test (e.g., k-s) where arbitrary data binning is not required.

We thank the reviewer for raising these issues. We agree on the fact that PV expression may be affected in an activity-dependent manner, as previously reported. However, we are unaware of any previous report interpreting the alterations in PV+ cells in light of activity changes in mouse models of DS. We followed the methodology of these reports in order to evaluate PV+ cell density in our mice.

Regarding the statistical methods, we agree on the fact that informing on the statistical tests used in each occasion would be important. However, due to the high number of statistical analyses performed, the test and the exact n- and p-values are properly stated in the supplementary tables instead of along the text, since this will make reading extremely demanding. In all statistical tests, corrections for multiple comparisons were automatically performed by the analysis software used (now indicated in page 19). Finally, regarding the analyses in Fig. 2a and c, we didn't perform k-s comparisons due to the fact that we had three distributions to compare. In contrast, we chose the bin analysis because of its simplicity and straightforward interpretation.

References for the point-by-point answers to the referees

Aroniadou-Anderjaska, V., Pidoplichko, V.I., Figueiredo, T.H., Almeida-Suhett, C.P., Prager, E.M., and Braga, M.F.M. (2012). Presynaptic facilitation of glutamate release in the basolateral amygdala: A mechanism for the anxiogenic and seizurogenic function of GluK1 receptors. *Neuroscience* 221, 157-169.

Best, T.K., Cramer, N.P., Chakrabarti, L., Haydar, T.F., and Galdzicki, Z. (2012). Dysfunctional hippocampal inhibition in the Ts65Dn mouse model of Down syndrome. *Experimental Neurology* 233, 749-757.

Bonfardin, V.D., Fossat, P., Theodosis, D.T., and Oliet, S.H. (2010). Glia-dependent switch of kainate receptor presynaptic action. *Journal of Neuroscience* 30, 985-995.

Braga, M.F.M., Aroniadou-Anderjaska, V., Li, H., and Rogawski, M.A. (2009). Topiramate Reduces Excitability in the Basolateral Amygdala by Selectively Inhibiting GluK1 (GluR5) Kainate Receptors on Interneurons and Positively Modulating GABAA Receptors on Principal Neurons. *Journal of Pharmacology and Experimental Therapeutics* 330, 558-566.

Braudeau, J., Delatour, B., Duchon, A., Pereira, P.L., Dauphinot, L., de Chaumont, F., Olivo-Marin, J.C., Dodd, R.H., Herault, Y., and Potier, M.C. (2011). Specific targeting of the GABA-A receptor alpha5 subtype by a selective inverse agonist restores cognitive deficits in Down syndrome mice. *Journal of Psychopharmacology* 25, 1030-1042.

Chakrabarti L, Galdzicki Z, Haydar TF. (2007) Defects in embryonic neurogenesis and initial synapse formation in the forebrain of the Ts65Dn mouse model of Down syndrome. *J Neurosci.* 27, 11483-95.

Chakrabarti, L., Best, T.K., Cramer, N.P., Carney, R.S., Isaac, J.T., Galdzicki, Z., and Haydar, T.F. (2010). Olig1 and Olig2 triplication causes developmental brain defects in Down syndrome. *Nature Neuroscience* 13, 927-934.

Christensen, J.K., Paternain, A.V., Selak, S., Ahring, P.K., and Lerma, J. (2004). A Mosaic of Functional Kainate Receptors in Hippocampal Interneurons. *Journal of Neuroscience* 24, 8986-8993.

Clark S, Schwalbe J, Stasko MR, Yarowsky PJ, Costa AC. (2006). Fluoxetine rescues deficient neurogenesis in hippocampus of the Ts65Dn mouse model for Down syndrome. *Exp Neurol.* 200, 256-61.

Contestabile, A., Magara, S., and Cancedda, L. (2017). The GABAergic Hypothesis for Cognitive Disabilities in Down Syndrome. *Frontiers in Cellular Neuroscience* 11, 54.

Corrales, A., Martínez, P., García, S., Vidal, V., García, E., Flórez, J., Sanchez-Barceló, E.J., Martínez-Cué, C., and Rueda, N. (2013). Long-term oral administration of melatonin improves spatial learning and memory and protects against cholinergic degeneration in middle-aged Ts65Dn mice, a model of Down syndrome. *Journal of Pineal Research* 54, 346-358.

Corrales A, Parisotto EB, Vidal V, García-Cerro S, Lantigua S, Diego M, Wilhem Filho D, Sanchez-Barceló EJ, Martínez-Cué C, Rueda N. (2017) Pre- and post-natal melatonin administration partially regulates brain oxidative stress but does not improve cognitive or histological alterations in the Ts65Dn mouse model of Down syndrome. *Behav Brain Res* 334,142-154.

Cossart, R., Tyzio, R., Dinocourt, C., Esclapez, M., Hirsch, J.C., Ben-Ari, Y., and Bernard, C. (2001). Presynaptic Kainate Receptors that Enhance the Release of GABA on CA1 Hippocampal Interneurons. *Neuron* 29, 497-508.

Costa, A., and Grybko, M.J. (2005). Deficits in hippocampal CA1 LTP induced by TBS but not HFS in the Ts65Dn mouse: A model of Down syndrome. *Neuroscience Letters* 382, 317-322.

Costa, A.C., Scott-McKean, J.J., and Stasko, M.R. (2008). Acute injections of the NMDA receptor antagonist memantine rescue performance deficits of the Ts65Dn mouse model of Down syndrome on a fear conditioning test. *Neuropsychopharmacology* 33, 1624-1632.

Cunha, R.A., Malva, J.O., and Ribeiro, J.A. (2000). Pertussis toxin prevents presynaptic inhibition by kainate receptors of rat hippocampal [3H]GABA release. *FEBS Letters* 469, 159-162.

Daw, M.I., Pelkey, K.A., Chittajallu, R., and McBain, C.J. (2010). Presynaptic Kainate Receptor Activation Preserves Asynchronous GABA Release Despite the Reduction in Synchronous Release from Hippocampal Cholecystinin Interneurons. *Journal of Neuroscience* 30, 11202-11209.

Fernandez, F., Morishita, W., Zuniga, E., Nguyen, J., Blank, M., Malenka, R.C., and Garner, C.C. (2007). Pharmacotherapy for cognitive impairment in a mouse model of Down syndrome. *Nature Neuroscience* 10, 411-413.

Fernandez, F., and Garner, C.C. (2007). Over-inhibition: a model for developmental intellectual disability. *Trends in Neurosciences* 30, 497-503.

García-Cerro, S., Martínez, P., Vidal, V., Corrales, A., Flórez, J., Vidal, R., Rueda, N., Arbonés, M.L., and Martínez-Cué, C. (2014). Overexpression of Dyrk1A Is Implicated in Several Cognitive, Electrophysiological and Neuromorphological Alterations Found in a Mouse Model of Down Syndrome. *PLoS ONE* 9. e106572.

García-Cerro, S., Rueda, N., Vidal, V., Lantigua, S., and Martínez-Cué, C. (2017). Normalizing the gene dosage of Dyrk1A in a mouse model of Down syndrome rescues several Alzheimer's disease phenotypes. *Neurobiology of Disease* 106, 76-88.

Granholm, A.C., Sanders, L.A., and Cnric, L.S. (2000). Loss of cholinergic phenotype in basal forebrain coincides with cognitive decline in a mouse model of Down's syndrome. *Experimental Neurology* 161, 647-663.

Hanson, J.E., Blank, M., Valenzuela, R.A., Garner, C.C., and Madison, D.V. (2007). The functional nature of synaptic circuitry is altered in area CA3 of the hippocampus in a mouse model of Down's syndrome. *Journal of Physiology* 579, 53-67.

Holtzman, D.M., Santucci, D., Kilbridge, J., Chua-Couzens, J., Fontana, D.J., Daniels, S.E., Johnson, R.M., Chen, K., Sun, Y., Carlson, E., et al. (1996). Developmental abnormalities and age-related neurodegeneration in a mouse model of Down syndrome. *Proc Natl Acad Sci U S A*. 93, 13333-8.

Ishihara, K., Amano, K., Takaki, E., Ebrahim, A.S., Shimohata, A., Shibasaki, N., Inoue, I., Takaki, M., Ueda, Y., Sago, H., et al. (2009). Increased lipid peroxidation in Down's syndrome mouse models. *Journal of Neurochemistry* 110, 1965-1976.

Ishihara, K., Amano, K., Takaki, E., Shimohata, A., Sago, H., Epstein, C.J., and Yamakawa, K. (2010). Enlarged brain ventricles and impaired neurogenesis in the Ts1Cje and Ts2Cje mouse models of Down syndrome. *Cerebral Cortex* 20, 1131-1143. Izquierdo, I., Furini, C.R., and Myskiw, J.C. (2016). Fear Memory. *Physiological Reviews* 96, 695-750.

Jiang, L., Xu, J., Nedergaard, M., and Kang, J. (2001). A Kainate Receptor Increases the Efficacy of GABAergic Synapses. *Neuron* 30, 503-513.

Jiang, Y., Rigoglioso, A., Peterhoff, C.M., Pawlik, M., Sato, Y., Bleiwas, C., Stavrides, P., Smiley, J.F., Ginsberg, S.D., Mathews, P.M., et al. (2016). Partial BACE1 reduction in a Down syndrome mouse model blocks Alzheimer-related endosomal anomalies and cholinergic neurodegeneration: role of APP-CTF. *Neurobiology of Aging* 39, 90-98.

Kaur, G., Sharma, A., Xu, W., Gerum, S., Alldred, M.J., Subbanna, S., Basavarajappa, B.S., Pawlik, M., Ohno, M., Ginsberg, S.D., et al. (2014). Glutamatergic transmission aberration: a major cause of behavioral deficits in a murine model of Down's syndrome. *Journal of Neuroscience* 34, 5099-5106.

Kleschevnikov, A.M., Belichenko, P.V., Villar, A.J., Epstein, C.J., Malenka, R.C., and Mobley, W.C. (2004). Hippocampal Long-Term Potentiation Suppressed by Increased Inhibition in the Ts65Dn Mouse, a Genetic Model of Down Syndrome. *Journal of Neuroscience* 24, 8153-8160.

Kleschevnikov, A.M., Belichenko, P.V., Faizi, M., Jacobs, L.F., Htun, K., Shamloo, M., and Mobley, W.C. (2012a). Deficits in Cognition and Synaptic Plasticity in a Mouse Model of Down Syndrome Ameliorated by GABAB Receptor Antagonists. *Journal of Neuroscience* 32, 9217-9227.

Kleschevnikov, A.M., Belichenko, P.V., Gall, J., George, L., Nosheny, R., Maloney, M.T., Salehi, A., and Mobley, W.C. (2012b). Increased efficiency of the GABAA and GABAB receptor-mediated neurotransmission in the Ts65Dn mouse model of Down syndrome. *Neurobiology of Disease* 45, 683-691.

Kleschevnikov, A.M., Yu, J., Kim, J., Lysenko, L.V., Zeng, Z., Yu, E.Y., and Mobley, W.C. (2017). Evidence that increased *Kcnj6* gene dose is necessary for deficits in behavior and dentate gyrus synaptic plasticity in the Ts65Dn mouse model of Down syndrome. *Neurobiology of Disease* 103, 1-10.

Kobayashi, K., Emson, P.C., Mountjoy, C.Q., Thornton, S.N., Lawson, D.E., and Mann, D.M. (1990). Cerebral cortical calbindin D28K and parvalbumin neurones in Down's syndrome. *Neuroscience Letters* 113, 17-22.

Leão, R.N., Mikulovic, S., Leão, K.E., Munguba, H., Gezelius, H., Enjin, A., Patra, K., Eriksson, A., Loew, L.M., Tort, A.B.L., et al. (2012). OLM interneurons differentially modulate CA3 and entorhinal inputs to hippocampal CA1 neurons. *Nature Neuroscience* 15, 1524-1530.

Lerma, J., and Marques, J.M. (2013). Kainate receptors in health and disease. *Neuron* 80, 292-311.

Martínez-Cué, C., Martínez, P., Rueda, N., Vidal, R., García, S., Vidal, V., Corrales, A., Montero, J.A., Pazos, Á., Flórez, J., et al. (2013). Reducing GABAA α 5 Receptor-Mediated Inhibition Rescues Functional and Neuromorphological Deficits in a Mouse Model of Down Syndrome. *Journal of Neuroscience* 33, 3953-3966.

Mulle, C., Sailer, A., Swanson, G.T., Brana, C., O'Gorman, S., Bettler, B., and Heinemann, S.F. (2000). Subunit composition of kainate receptors in hippocampal interneurons. *Neuron* 28, 475-484.

Parrini M, Ghezzi D, Deidda G, Medrihan L, Castroflorio E, Alberti M, Baldelli P, Cancedda L, Contestabile A. Aerobic exercise and a BDNF-mimetic therapy rescue learning and memory in a mouse model of Down syndrome. *Sci Rep.* 7, 16825.

Paternain, A.V., Herrera, M.a.T., Nieto, A.M., and Lerma, J. (2000). GluR5 and GluR6 Kainate Receptor Subunits Coexist in Hippocampal Neurons and Coassemble to Form Functional Receptors. *Journal of Neuroscience* 20, 196-205.

Reeves, R.H., Irving, N.G., Moran, T.H., Wohn, A., Kitt, C., Sisodia, S.S., Schmidt, C., Bronson, R.T., and Davisson, M.T. (1995). A mouse model for Down syndrome exhibits learning and behaviour deficits. *Nature Genetics* 11, 177-184.

Rodríguez-Moreno, A., and Lerma, J. (1998). Kainate Receptor Modulation of GABA Release Involves a Metabotropic Function. *Neuron* 20, 1211-1218.

Ross, M.H., Galaburda, A.M., and Kemper, T.L. (1984). Down's syndrome: Is there a decreased population of neurons? *Neurology* 34, 909-916.

Shichiri M, Yoshida Y, Ishida N, Hagihara Y, Iwahashi H, Tamai H, Niki E. (2011). α -Tocopherol suppresses lipid peroxidation and behavioral and cognitive impairments in the Ts65Dn mouse model of Down syndrome. *Free Radic Biol Med.* 50, 1801-11.

Siarey, R.J., Stoll, J., Rapoport, S.I., and Galdzicki, Z. (1997). Altered long-term potentiation in the young and old Ts65Dn mouse, a model for Down Syndrome. *Neuropharmacology* 36, 1549-1554.

Siwani, S., França, A., Mikulovic, S., Reis, A., Hilscher, M.M., Edwards, S.J., Leão, R.N., Tort, A., and Kullander, K. (2018). OLM α 2 Cells Bidirectionally Modulate Learning. *Neuron* 99, 404-412.

Stagni, F., Magistretti, J., Guidi, S., Ciani, E., Mangano, C., Calzà, L., and Bartesaghi, R. (2013). Pharmacotherapy with Fluoxetine Restores Functional Connectivity from the Dentate Gyrus to Field CA3 in the Ts65Dn Mouse Model of Down Syndrome. *PLoS ONE* 8, e61689.

Turi GF, Li WK, Chavlis S, Pandi I, O'Hare J, Priestley JB, Grosmark AD, Liao Z, Ladow M, Zhang JF, Zelman BV, Poirazi P, Losonczy A. Vasoactive Intestinal Polypeptide-Expressing Interneurons in the Hippocampus Support Goal-Oriented Spatial Learning. *Neuron.* 101, 1150-1165.e8.

Villar, A.J., Belichenko, P.V., Gillespie, A.M., Kozy, H.M., Mobley, W.C., and Epstein, C.J. (2005). Identification and characterization of a new Down syndrome model, Ts[Rb(12.1716)]2Cje, resulting from a spontaneous Robertsonian fusion between T(1716)65Dn and mouse Chromosome 12. *Mammalian Genome* 16, 79-90.

Wyeth, M.S., Pelkey, K.A., Yuan, X., Vargish, G., Johnston, A.D., Hunt, S., Fang, C., Abebe, D., Mahadevan, V., Fisahn, A., et al. (2017). Neto Auxiliary Subunits Regulate Interneuron Somatodendritic and Presynaptic Kainate Receptors to Control Network Inhibition. *Cell Reports* 20, 2156-2168.

Zorrilla de Martin, J., Delabar, J.-M., Bacci, A., and Potier, M.-C. (2018). GABAergic overinhibition, a promising hypothesis for cognitive deficits in Down syndrome. *Free Radical Biology and Medicine* 114, 33-39.

REVIEWERS' COMMENTS:

Reviewer #1 (Remarks to the Author):

The authors have adequately answered to most of my questions and issues. The overall quality of the manuscript has improved.

Reviewer #2 (Remarks to the Author):

The authors have addressed all my major concerns with their comments, additional data and text changes.

The only outstanding point is the inclusion of the correlation plots (now Figure S12), which I would continue to recommend be removed from the paper, however I will leave this up to the discretion of the editor. Overall I find this to be an interesting study that will be of value to the field.

One minor error on p.3, "miss-expression" should be misexpression

Reviewer #3 (Remarks to the Author):

There is very little for me to write, as the authors responded to almost none of my previous concerns. Thus, they remain as before. Highlighted briefly:

1. Altered pain sensitivity and rotarod fall latency are hardly a reasonable assay of altered sensory function that might contribute to the phenotype (can they see? can they hear? can they smell?).
2. The authors did nothing to address the almost complete lack of consideration for other mechanisms that would alter mIPSCs beyond dendritic location (e.g., receptor subunit composition or altered kinetics of presynaptic release). Indeed, they seem to confuse miniature events with asynchronous release. These are not the same, and the former does not measure the latter.
3. I still feel the LTP data are out of place in this story. The fact that the result is important does not make it peripheral to the present manuscript.
4. Statistical problems are largely unaddressed. The authors do not seem to realize that arbitrary binning of data is not a valid analysis approach. Additionally, they do not even name the multiple comparisons correction, only stating that the software does it automatically. This does not inspire confidence in the overall statistics.
5. There is no response to my concern that measuring PV expression is not "cell counting", as levels of PV are highly activity dependent. Thus, changes in PV "counts" may have no relation to the number of cells that are actually canonical PV interneurons.

Point by point answers to reviewers' comments.

REVIEWERS' COMMENTS:

We thank the reviewers for their thoughtful comments. We believe that, overall, they have greatly improved the quality of the manuscript. Below we provide a point-by-point answer to these comments.

Reviewer #1 (Remarks to the Author):

The authors have adequately answered to most of my questions and issues. The overall quality of the manuscript has improved.

We are glad for having successfully answered the reviewer's questions and thank the reviewer for the help in improving the quality of the manuscript.

Reviewer #2 (Remarks to the Author):

The authors have addressed all my major concerns with their comments, additional data and text changes.

We are also happy for having complied with the reviewer's concerns and thank him/her for the help in the process.

The only outstanding point is the inclusion of the correlation plots (now Figure S12), which I would continue to recommend be removed from the paper, however I will leave this up to the discretion of the editor. Overall I find this to be an interesting study that will be of value to the field.

We still believe that the correlations provided in the Supplementary Fig. 12 improve the quality of the work, as they provide a plausible explanation for the observed phenotypes in the fear conditioning test. Indeed, the existence of correlations between apparently unrelated behaviours suggest some caution at the time of making interpretations from different behavioral paradigms in this and other studies. We thank the Editor for allowing the figure to remain.

One minor error on p.3, "miss-expression" should be misexpression

We thank the reviewer for highlighting this typo and we have changed the text accordingly.

Reviewer #3 (Remarks to the Author):

There is very little for me to write, as the authors responded to almost none of my previous concerns. Thus, they remain as before. Highlighted briefly:

We thank the reviewer for the thoughtful comments. We try to provide meaningful answers to his/her points.

1. Altered pain sensitivity and rotarod fall latency are hardly a reasonable assay of altered sensory function that might contribute to the phenotype (can they see? can they hear? can they smell?).

We thank the reviewer for raising this point. It is a convenient question to ask whether the animals have altered sensory function. According to the previous concern raised by the referee –regarding motor function and pain sensitivity- we have performed the rotarod and hot plate tests, finding no differences across genotypes. Regarding the new issue dealing with sensory function, we did not find any gross phenotype. Actually, during NOL and NOR tasks, mice are guided by visual cues and sniff the objects as a way to interact with them, suggesting that sensory function is not altered.

2. The authors did nothing to address the almost complete lack of consideration for other mechanisms that would alter mIPSCs beyond dendritic location (e.g., receptor subunit composition or altered kinetics of presynaptic release). Indeed, they seem to confuse miniature events with asynchronous release. These are not the same, and the former does not measure the latter.

We thank the reviewer for highlighting this point. Although a change in the weight of asynchronous vs synchronous release would affect evoked IPSC, this is unlikely to affect the kinetics of single events. Therefore, our assumption of the origin of mIPSC according to their kinetics stands. Conceptually, asynchronous events and a miniature events are identical (Kaesser and Regehr, *Annu Rev Physiol.* 2014; 76: 333–363). In any case, we have included a statement in the manuscript about a plausible mechanism (i.e. altered synapse density) that may also participate in alterations in mIPSCs frequency besides the one we have demonstrated (i.e. unbalanced dendritic inhibition caused by GluK1 dependent alterations in GABA release probability).

3. I still feel the LTP data are out of place in this story. The fact that the result is important does not make it peripheral to the present manuscript.

We still believe that the inclusion of the LTP data in the paper is important due to two issues. First, it discards the participation of *Grik1* triplication in the appearance of the classical LTP deficits found in mouse models of DS. Second, it forces to reconsider the long-established idea that spatial memory alterations are exclusively due to LTP deficits in the hippocampus. However, in order to further integrate the information provided in the LTP experiments, we have moved the synaptic plasticity figure and the text section ahead to the mIPSCs analysis. We believe that this change improves the narrative of the article, making it easier to understand the reason for performing the LTP experiments.

4. Statistical problems are largely unaddressed. The authors do not seem to realize that arbitrary binning of data is not a valid analysis approach. Additionally, they do not even name the multiple comparisons correction, only stating that the software does it automatically. This does not inspire confidence in the overall statistics.

We thank the reviewer for this point. However, we disagree regarding the reviewer's idea on the use of data binning as a valid analytic approach. We performed such data binning as a way to specifically evaluate the effect of *Grik1* triplication in the events (m/sIPSCs) originated in different sections of the dendrite. We believe data binning is a totally valid approach for this purpose although, in any case, we have further confirmed the conclusion from this in-depth analysis of m/sIPSCs by assessing PPR of eIPSCs in different segments of the somatodendritic axis of CA1 pyramidal cells. Regarding multiple comparisons, we state that corrections for multiple comparisons were automatically performed by the tests we used in the Statistics subsection of the Methods (page 20).

5. There is no response to my concern that measuring PV expression is not "cell counting", as levels of PV are highly activity dependent. Thus, changes in PV "counts" may have no relation to the number of cells that are actually canonical PV interneurons.

We agree with the reviewer regarding the partial activity-dependence of PV expression. However, although we cannot discard that such activity dependence may influence interneuron counts in our mice, we don't believe this is critical. No previous study has reported any difference in activity-dependence of PV expression in mouse models of DS (even though multiple laboratories have explored PV+ neuron densities in these models). Besides, in our study, the PV+ cell density was not altered by any change in environmental conditions, suggesting that activity levels played negligible roles in PV expression. In any case, we have included a statement highlighting this possibility in the manuscript.